# Understanding the spread of agriculture in the Western Mediterranean (6th-3rd millennia BC) with Machine Learning tools

Maria Elena Castiello [1,2] ✉, Emmanuele Russo[3], Héctor Martínez-Grau[4], Ana Jesus[4], Georgina Prats [4,5] & Ferran Antolín [2,4]

The first Neolithic farmers arrived in the Western Mediterranean area from the East. They established settlements in coastal areas and over time migrated to new environments, adapting to changing ecological and climatic conditions. While farming practices and settlements in the Western Mediterranean differ greatly from those known in the Eastern Mediterranean and central Europe, the extent to which these differences are connected to the local environment and climate is unclear. Here, we tackle this question by compiling data and proxies at a superregional and multi-scale level, including archaeobotanical information, radiocarbon dates and paleoclimatic models, then applying a machine learning approach to investigate the impact of ecological and climatic constraints on the first Neolithic humans and crops. This approach facilitates calculating the pace of spread of farming in the Western Mediterranean area, modelling and estimating the potential areas suitable for settlement location, and discriminating distinct types of crop cultivation under changing climatic conditions that characterized the period 5900 – 2300 cal. BC. The results of this study shed light onto the past climate variability and its influence on human distribution in the Western Mediterranean area, but also discriminate sensitive parameters for successful agricultural practices.

The spread of early farmers from SW Asia towards Western Europe is a testament to the capacity of Neolithic farmers to succeed in many different environments, substrates and climatic regions. This success is possible partly thanks to the large variety of crops available at the beginning of this process, the intensive (garden-plot-type) and mixed (closely integrated with animal management) nature of farming[1], but also the important role of wild plants in the diet[2–4]. There is abundant literature reviewing how the spread of these populations was only possible by focusing on the crops better adapted to the new climatic conditions[5–8]. However, so far no research has quantitatively assessed the ecological niche of these crops in the past, and the degree to which these farming communities and the crops they cultivated were

constrained by or adapted to climate change events that potentially happened during the Neolithic period (between 5900 and 2300 cal. BC)[9].

The North-west Mediterranean region and the western Alpine Foreland (current Switzerland) is one of the best-investigated areas regarding settlement patterns and agricultural practices in the Western Mediterranean and also one of the regions that have seen a greater improvement of the datasets in recent times. In comparison to central Europe, the area is less intensively known[10], particularly for the Early Neolithic (5900–4500 BC, our Phase 01 and Phase 02), but sites are well radiocarbon dated, partly due to the establishment of radiocarbon dates as proxies to understand the Neolithisation process[11,12] and partly

[1]Institut d'archéologie et des sciences de l'antiquité, University of Lausanne, Lausanne, Switzerland. [2]Division of Natural Sciences, German Archaeological Institute, Berlin, Germany. [3]Institute for Atmospheric and Climate Science, ETH Zurich, Zurich, Switzerland. [4]Integrative Prehistory and Archaeological Science, Department of Environmental Sciences, University of Basel, Basel, Switzerland. [5]ARQHISTEC, Grup d'Investigació Prehistòrica (GIP-UdL), Departament de Geografia, Història i Història de l'Art, Universitat de Lleida, Lleida, Spain. ✉e-mail: MariaElena.Castiello@unil.ch

because of the long stratigraphies preserved in cave sites requiring accurate dating and modelling[13], but particularly thanks to dendrochronological dating of waterlogged sites. The Middle (4500–3500/3300 BC, our Phase 03) and Late Neolithic (3500/3300–2300 BC, our Phase 04) periods are very well researched, particularly if we consider the large amount of high-quality data coming from the pile-dwellings in current Swiss territory and the abundant research carried out in some of the other areas, partly connected to rescue archaeology interventions as human impact on the landscape of the region increases[14–16].

It is commonly agreed that the first farmers that got to the region spread along the coasts since new populations arrived through navigation, and progressively moved inland. Recent work has questioned this assumption[13]. Manen and others highlighted that early Neolithic settlements seem to have optimized adapting capacities to dwell in different environments and different types of sites: open-air sites, cave sites, pile dwellings, etc.[10]. Whether this observation translates in a particularly diverse niche or only in diverse topographical locations with similar conditions is unclear. Between the 5th and 3rd millennia strong networks develop. In this context, the settling of farmers in the Swiss Plateau (and around lakes in the Jura region) takes place, probably connected to a spread of populations from the South[12,17]. There is abundant evidence of changing technologies due to internal dynamics and external influences, as well as of the exchange of prestige goods and even small-scale migrations, particularly at the end of the Neolithic[18]. According to available archaeological evidence, migrating individuals did not necessarily move in large population waves, and they integrated into existing villages, which is well documented, for instance, in the Swiss pile dwellings[19]. Considering this, for the moment, changes in niche amplitude should not be understood as evidence of the arrival of new groups that prefer other locations. It is our hypothesis, that changes in the niche breadth imply economic changes that may or may not be connected to climatic changes. Authors observe an important use of middle/low mountain ranges in the 6th and 5th millennia BC[20,21], but the interpretations of this phenomenon differ, either as evidence of an economic specialization or of permanent occupations at middle altitudes. Higher mountain ranges would be seasonally targeted in the 4th and 3rd millennia BC[20,22].

Actually, a long-term analysis of the changing niches in the Neolithic has not been evaluated in combination with agricultural practices. This is an important research gap considering that Early Neolithic societies in the area are poorly stratified and site location will probably be mostly driven by environmental factors as well as social networks. The use of niche modelling techniques is generally not new, and our efforts here align with the most recent studies in archaeology[23–27] trying to assess the responses of human societies to climate variability, making use of the latest high-resolution climate model results. 'Habitat suitability' (HS) is (as detailed in Methods, Habitat Suitability and Niche models construction, following the work of Braunisch and others[28]) an inference based on extrapolation from archaeological site distributions, and not on a priori arguments about the suitability of different landscapes. It has the implicit assumption that archaeological sites must have been located in areas that are suitable for human habitation at a given time. Blinkhorn and others[29] have indeed tried to infer human behaviour from the archaeological record of the Late Pleistocene, while Banks and others[30] adopted more specific environmental and cultural niche modelling techniques to conclude that environmental factors did have an influence on the predisposing occupation of regions most suited to specific cultural adaptations for the prehistoric farmers. Such first quantitative attempts, and the process of exploring and explaining where and why Neolithic populations occurred and settled, and to which extent people's lives were already affected by climatic factors and constraints has rightly become a central focus of debate for an increasing number of archaeological studies, concurrently with the more pressing concern and challenge of

the global climate crisis[31,32]. Computational and quantitative modelling techniques come in hand and can be of greatest benefit for archaeologists trying to address large-scale events connected with relevant modern challenges. When considering the greatest amount of data and information available today, especially when approaching the complex phenomena of the spread of agriculture that includes dozens of countries, we realize that computer tools and more advanced statistical methods are essential to combine and integrate multiple proxies and databases at different spatiotemporal scales, to test and validate several hypotheses indeed formulated to reconstruct the past.

Previous research has focused on the spread or on the understanding of expansions and declines of certain archaeological cultures. We nevertheless do not advocate for deterministic positions aiming to explain the disappearance of sites with certain pottery decorations. We are more interested in the relationship between site location and crop diversity at a given time and place since crops have a more direct relationship with weather and climate.

Among the crops available to early farmers arriving into current mainland Greece from SW Asia, we can consider naked wheat (*Triticum aestivum/durum*), emmer (*Triticum dicoccon*), einkorn (*Triticum monococcum*), Timopheev's wheat (*Triticum timopheevii*), hulled barley (*Hordeum distichon/vulgare*), naked barley (*Hordeum vulgare* var. *nudum*), lentil (*Lens culinaris*), pea (*Pisum sativum*), broad bean (*Vicia faba*), chickpea (*Cicer arietinum*), bitter vetch (*Vicia ervilia*) and flax (*Linum usitatissimum*). The characteristics of the different cultivated plants and their uses differ, as highlighted by numerous authors[33–39]. Einkorn grows well on poor soils and in cold areas and tolerates wet climates. It is mostly used to produce bulgur and similar products instead of bread-like foodstuffs. Emmer is slightly more drought-resistant than einkorn and a bit more demanding on soil quality. It is also more productive. Timopheev's wheat is the most wet-tolerant cereal of all and is also resistant to many cereal plant pathogens[40]. Barleys have a shorter growing period, and this makes them more adaptable to arid conditions, while they also grow well on poor soils. They can be turned into flour but with a low starch content, hence mostly used in porridge soups as roasted grains or added to other flours to produce bread.

Chickpea[41] is one of the crops that is first abandoned as farmers started spreading towards the European continent, although there are a few occurrences in the Iberian Peninsula[42]. Naked wheat and broad bean only seem to spread along the Mediterranean coast, but not towards central Europe until later chronologies. One final crop, opium poppy (*Papaver somniferum*) seems to have been taken into cultivation in the Western Mediterranean and then spread towards central Europe together with naked wheat[43–45]. The reason for the early abandonment of naked wheat and chickpea by farmers entering the Carpathian Basin may be due to the continental climatic conditions, which made it more difficult for these crops to thrive. Conversely, in the Western Mediterranean, naked wheat was a very important crop, and it eventually overtook the role of hulled wheat in the economy[7,33]. After the first spread of farming, further changes in the crop spectrum occurred in different regions. For instance, in the Western Mediterranean, according to previous research[46–48], a virtual replacement of naked wheat by einkorn, emmer and Timopheev's wheat occurred at ca. 4000 BC. As crops expanded from the Western Mediterranean coast towards the Alpine area (both southwest and northwest)[17] new climates had to be faced. Although this condition should have posed similar challenges to early farmers than the expansion towards the northern Balkans and the Hungarian plain, it seemed to not have affected their crop choice (with the great importance of naked wheat, naked barley and opium poppy, similar to other Mediterranean sites)[35]. Previous approaches to the reconstruction of ecological niches of Neolithic cultures[30] interpreted the potential differences in the ecological niches of early farming groups in central and southern Europe as deriving from the crops they were growing. While this may be

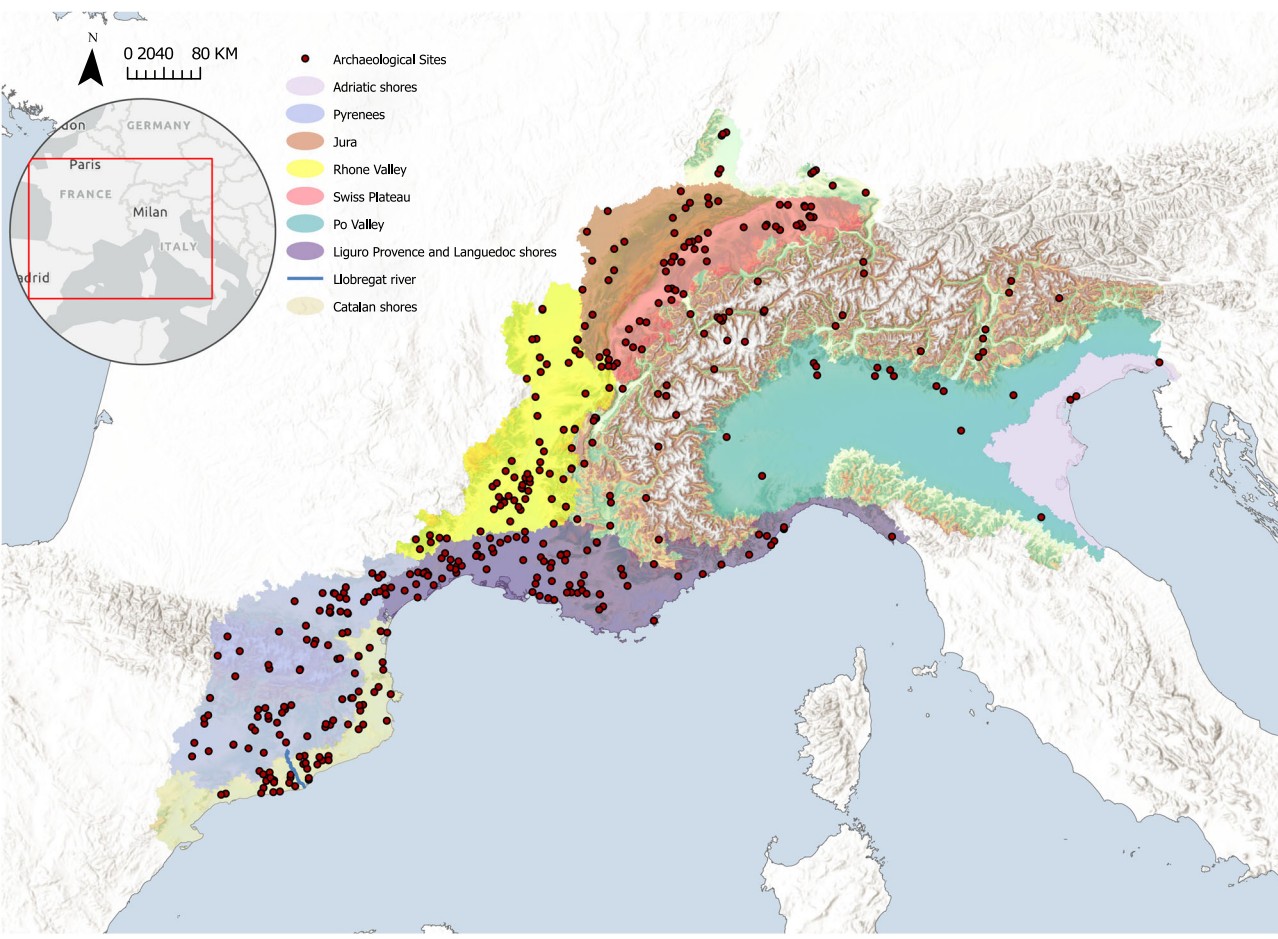

**Fig. 1 | Study Area.** The study area of the Western Mediterranean with the archaeological sites (red dots) and the ecologically diverse regions highlighted in different colours.

partially true, it is even more interesting to observe how farming evolved in well investigated regions and whether the niche of those crops actually expanded or reduced with time.

In this study, we propose an innovative multi-proxy and inter-disciplinary approach based on Machine learning (ML) algorithms, namely Random Forest (RF) and Maximum Entropy (MaxEnt) to characterize and quantify the mechanisms and the impact of climatic and environmental factors on early farmer settlement preferences and crop choices over the Neolithic period in the North-west Mediterranean region and Switzerland. We coupled a database consisting of a total of 3416 geo-referenced archaeological sites, with associated radiocarbon dates (calibrated using OxCal v. 4.4.2 and the atmospheric curve IntCal20) from *AgriChange_14Cdatabase*[49] and crop occurrences in archaeobotanical analyses obtained within the AgriChange project[50]. The study area (Fig. 1) of this research amounts in total to almost 310,000 km² of surface and offers a wide topographic and bioclimatic diversity, furthermore with ecologically diverse regions. By coupling an increased amount of published heterogeneous data and proxies at a superregional and multi-scale level, we demonstrate that climatic fluctuations, such as dropping or rising temperatures, increasing precipitation or decreasing seasonality values, were tightly intertwined with the history and distribution of the early farmers and especially their coping agricultural strategies.

Here, we show the extent to which the changes observed in crop distribution were climatically driven in this particular study area, characterizing expanding crop niches and climatic conditions across past chronologies, by coupling paleoclimatic data from the most recent high-resolution climate model results (CHELSA TraCE21k

dataset from Karger et al.[51]; see "Methods" section), with crop occur-rences. The main premise of the paper is that choices regarding site location and crop cultivation made in periods of climatic change most likely reflect the adaptive strategies of prehistoric populations.

The aims of the paper are to (1) Determine whether human and crop niches changed over time in the Neolithic; (2) Establish if new niches followed climate changes but remained stable in character or if they changed in climatic/landscape factors; (3) Observe if new niches were tied to the adoption of new crops or if new crops were adopted within stable niches; (4) Generate maps of potentially suitable areas of early farmers' settlements; (5) Assess the capacity of these techniques to address key archaeological and archaeobotanical questions.

## Results

### Paleoclimate and environmental envelope

To evaluate Neolithic farmers and crop niches, we first provided a rapid characterization of past climate variability. We examined possi-ble patterns in downscaled annual precipitation and temperature values with the CHELSA-TraCE21k dataset[51] (for a complete overview of the temporal evolution of the CHELSA paleoclimatic variables see Supplementary Figs. 1 and 2) which has yet to be used more widely in studies modelling archaeological or archaeobotanical occurrences for the time frame selected here, and we could thus observe important variability in the Annual Mean Temperature (Bio01) and the Tem-perature Seasonality (Bio04) values, especially in correspondence and between the four Phases (see the vertical dashed grey lines in Fig. 2) in which the available archaeological and archaeobotanical information was organized following previous work[49] (see Supplementary Table 1

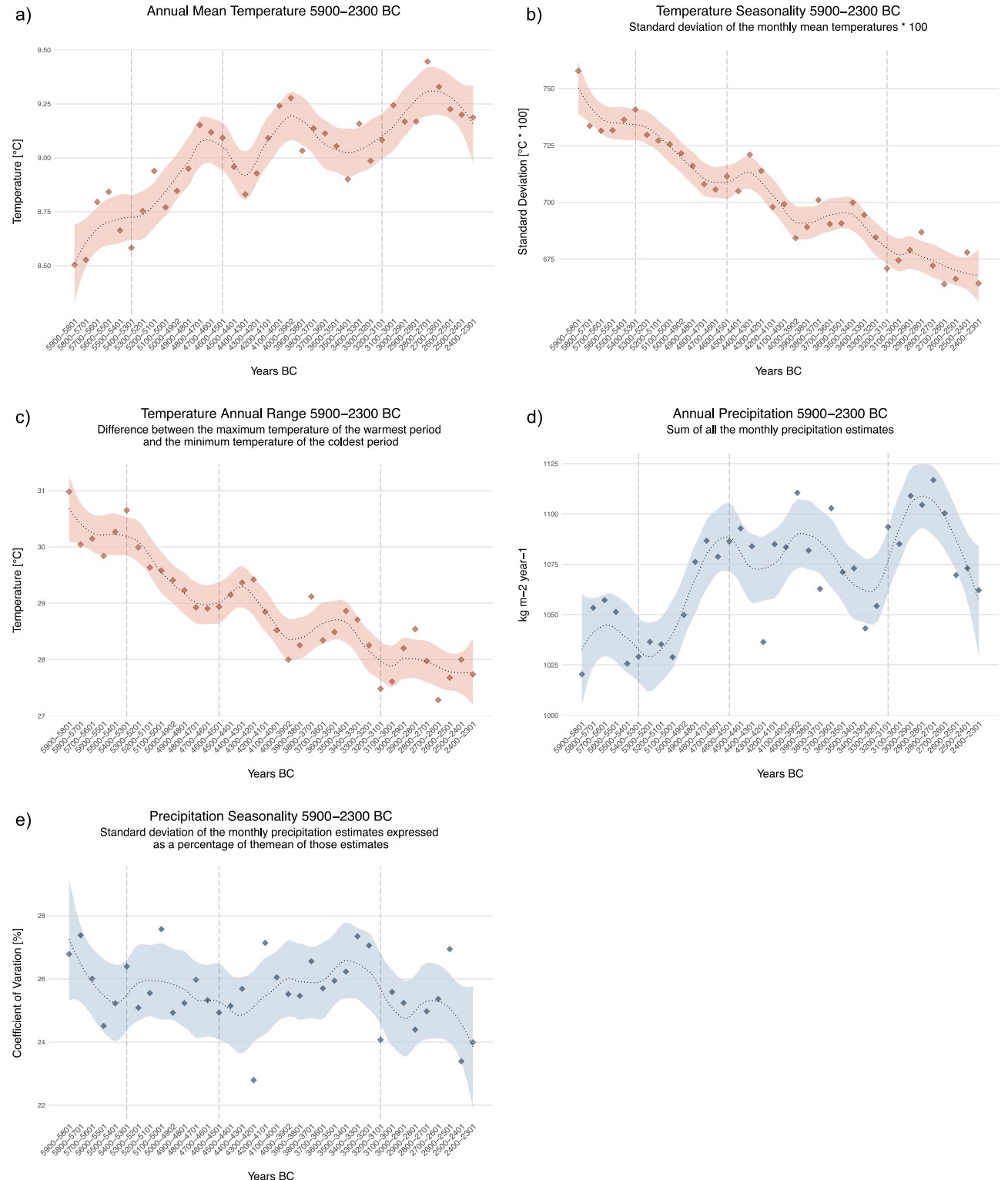

**Fig. 2 | Modelled climatic oscillation in the study area over the period 5900-2300 cal. BC** (derived from CHELSA-TraCE21k dataset). Evolution of Annual Mean temperature (**a**), Temperature Seasonality (**b**), Temperature Annual Range (**c**), Annual Precipitation (**d**) and Precipitation Seasonality (**e**) over the entire study area and period. The points represent the calculated mean value over the entire study area for each 100-year step. The dotted line represents the loess smooth with a span of 0.3 and the filled area the 95% confidence level interval (*geom_smooth* function from *ggplot2* package in *R*). The vertical dashed lines indicate the limits of our 4 chrono-phases. Source data can be found at https://doi.org/10.5281/zenodo. 14253277.

and Methods, Archaeological and Archaeobotanical data). These four Phases are based on crop dynamics and divide our study period according to the changing dominance of crop assemblages (for more details about the division into four Phases, see Supplementary Note 1).

In general terms, for the study area and the full period, a general increase in Annual Mean Temperature (Fig. 2a) is paired with a decrease in Temperature Seasonality (Fig. 2b) and in the Temperature Annual Range (Fig. 2c), reflecting an evolution towards a warmer and

less extreme climate for the Western Mediterranean area. Likewise, the Annual Precipitation increases over time while the Precipitation Seasonality remains generally constant, showing a less pronounced trend towards lower seasonality (Fig. 2d, e). Summers became cooler and wetter while winters became warmer with variable amounts of precipitation.

Considering these results, we would hypothesize that we should first find crops that benefit from stronger seasonality and lower temperatures, and there could be a progressive shift to crops that perhaps do not need such strong seasonality and are better prepared to withstand higher temperatures. We could also expect a spread of settlement locations to previously less favourable areas, such as higher altitudes, given a general trend to reduction of seasonality. Considering the Annual Precipitation, we observe a very dry phase around 5300-5200 BC, which coincides with the change from Phase 01 to Phase 02, and a very wet period in Phase 04. We could, in this sense hypothesize the appearance of highly drought-resistant crops in Phase 02 and wet-tolerant crops in Phase 04. We should also expect the appearance of wells in the driest phases, as a phenomenon known in other regions[52], but the evidence of wells is still sparse in our study region, perhaps due to taphonomic issues.

Given these general conditions, we then examined, more specifically, the ecological envelope for the settlements classified *per* type and phase (see Fig. 3 below; Supplementary Table 2and Supplementary Fig. 3a–d). Over the entire period, we found that the majority of sites were located in warm environments with an Annual Mean Temperature between ca. 10 and 15 °C (Fig. 3a) and Annual Precipitation values no higher than 1250 mm per year. They are especially clustered around 750 mm (Fig. 3b). A large portion of "open air" sites persistently occupied the areas in proximity to the main lakes and main rivers, especially during Phase 03 (Fig. 3c, d). Lower altitudes (below 700 m.a.s.l.) and gentler slopes were preferred by most of the sites (Fig. 3e, f; the complete list of these outputs can be found in Supplementary Fig. 3a–d).

## The Neolithic farmers' niche

We developed two HS models based on two different ML algorithms: RF and MaxEnt, using paleoclimatic and environmental variables as predictor inputs to produce maps of suitable areas for Neolithic farmers' occupation, one for each of the Phases identified (Figs. 4 and 5). These maps with high (purple) and low (green) values show that suitable areas for Neolithic settlements changed considerably over time. In particular, the maps produced by the RF algorithm (Fig. 4a–d) point to a significant change between 5900-4500 BC (Phase 01 and 02) and 4499–2300 BC (Phase 03 and 04). Looking closely at Fig. 4a, during the first phase (5900–5300 BC), the most suitable areas are essentially distributed along the Liguro-Provence and Languedoc shores (Northern Mediterranean shores), the Llobregat river mouth (Southern Mediterranean shores) and the Pyrenees. Consistent high suitability is also observed in the area of the Po valley and along the Adriatic shores until the end of the second phase (4500 BC) (Fig. 4b). Starting with Phase 03 (4499–3100 BC) in Fig. 4c, the most suitable areas seem to shift towards the inland, the lower-course of the Rhone valley, the Swiss Plateau and the main Swiss lakes, as well as the Jura region, thus abandoning the Mediterranean shores (Fig. 4d). The Italian Peninsula as well as the Alpine regions present very few suitable areas during these two more recent periods.

The suitability maps produced using MaxEnt (Fig. 5) show some differences but also some similarities in many cases with the maps produced using RF. During Phase 01 (Fig. 5a), there is an absence of suitable areas in the Po valley, the Italian Peninsula, the Swiss Plateau and the higher course of the Rhone valley. With Phase 02 (Fig. 5b), the map shows again higher suitability along the Po valley, along the Pyrenees and around Lake Garda, but less suitable areas over Liguro-Provence and Languedoc shores. During Phase 03 (4499–3100 BC),

higher probabilities are found around the alpine region, the Pyrenees and the lower Rhone valley. The highest suitability scores are more widely distributed over the landscape during the last phase (Fig. 5d), and reach further inland, away from the coast, compared to earlier phases. A shift that is supported both by the MaxEnt and the RF models.

The significance of each variable used in the two models is presented as *Variable Importance Ranking* (Supplementary Fig. 4) and *Partial Dependence plots* (Supplementary Fig. 5a–d) for RF. The *Variable Importance Ranking* for RF (using Mean Decrease Accuracy) is a statistical measure computed by looking at how much removing a variable decreases the model accuracy[53]. The *Partial Dependence plots* are a graphical indication of the influence (or marginal effect) of the specific class/range of values on the computed probability of site location[54].

For MaxEnt, the *Response Plots* (Supplementary Fig. 6a–d) indicate the role of a particular variable depicted as a response curve showing the predicted relative occurrence rate (suitable areas) against the values of that predictor (variable). As stated by Hong and others[53], "a key advantage of variable importance measures, as compared to univariate screening methods, is that they cover the impact of each predictor variable individually as well as in multivariate interactions with other predictor variables".

As a result of the RF computations, the Mean Temperature Diurnal Range (Bio02) and the distance to the main lakes (D_Lakes) are the two most important variables in predicting high suitability areas during the first Phase 01, with a peak of suitability prediction at a Temperature Diurnal Range of 6 °C and far away from the main lakes above 100 km. During Phase 02, Bio02 is associated with the elevation variable (DEM), and together they represent the two most important factors in defining the best ecological settings. Similar to Phase 01, the peak of suitability prediction of Bio02 is maintained between 6 °C and 8 °C. Regarding elevation, the highest positive dependence is shown below 500 m.a.s.l. During Phase 03 and 04, the distance to the main lakes plays the most relevant role in predicting the highest suitability areas, with the highest suitability in close proximity to them, associated with the Mean Temperature of Wettest Quarter (Bio08) during Phase 03 with values over 10 °C and to elevation during Phase 04, with positive values between 200 and 700 m.a.s.l. (see Supplementary Figs. 4 and 5a–d).

While MaxEnt provides more generalized response curves, the variable *Response Plots* show very similar results for these variables. The highest prediction appears again at a Mean Temperature Diurnal Range (Bio02) of 6 °C and gently decreases with increasing Bio02 values (during Phase 01 and 02). Similarly, the peak in prediction lies below 500 m.a.s.l. in Phase 02 with high values (> 0.5) up to 700 m.a.s.l. in Phase 04. While in Phase 01, the highest predictions correspond to a distance to lakes between 100 and 150 km, in Phase 03 and 04 they lie in close proximity to them (see Supplementary Fig. 6a–d).

The shift further inland and towards higher altitudes observed above when comparing the suitability maps computed by the two models has been also statistically inspected as the distribution of the predicted most suitable areas (with suitability ≥ 0.75) over the climatic and environmental features. The RF model shows that between 5900 and 4500 BC, the most suitable areas are those with an Annual Mean Temperature (Bio01) around 13 °C and mainly located at very low altitudes (with median values at 193 for Phase 01 and 210 m.a.s.l. for Phase 02 and a peak in the density of occurrences below these values). These preferences changed with the onset of Phase 03, when the best areas spread over a larger range both considering altitude and mean annual temperature. Median altitudes increase to 323 m.a.s.l (Phase 03) and 392 m.a.s.l. (Phase 04) with a mean annual temperature showing much larger ranges than in the previous phases (see Fig. 6a, b). The MaxEnt model shows very similar predictions, with a peak of high suitability areas at very low altitudes during Phase 01 and

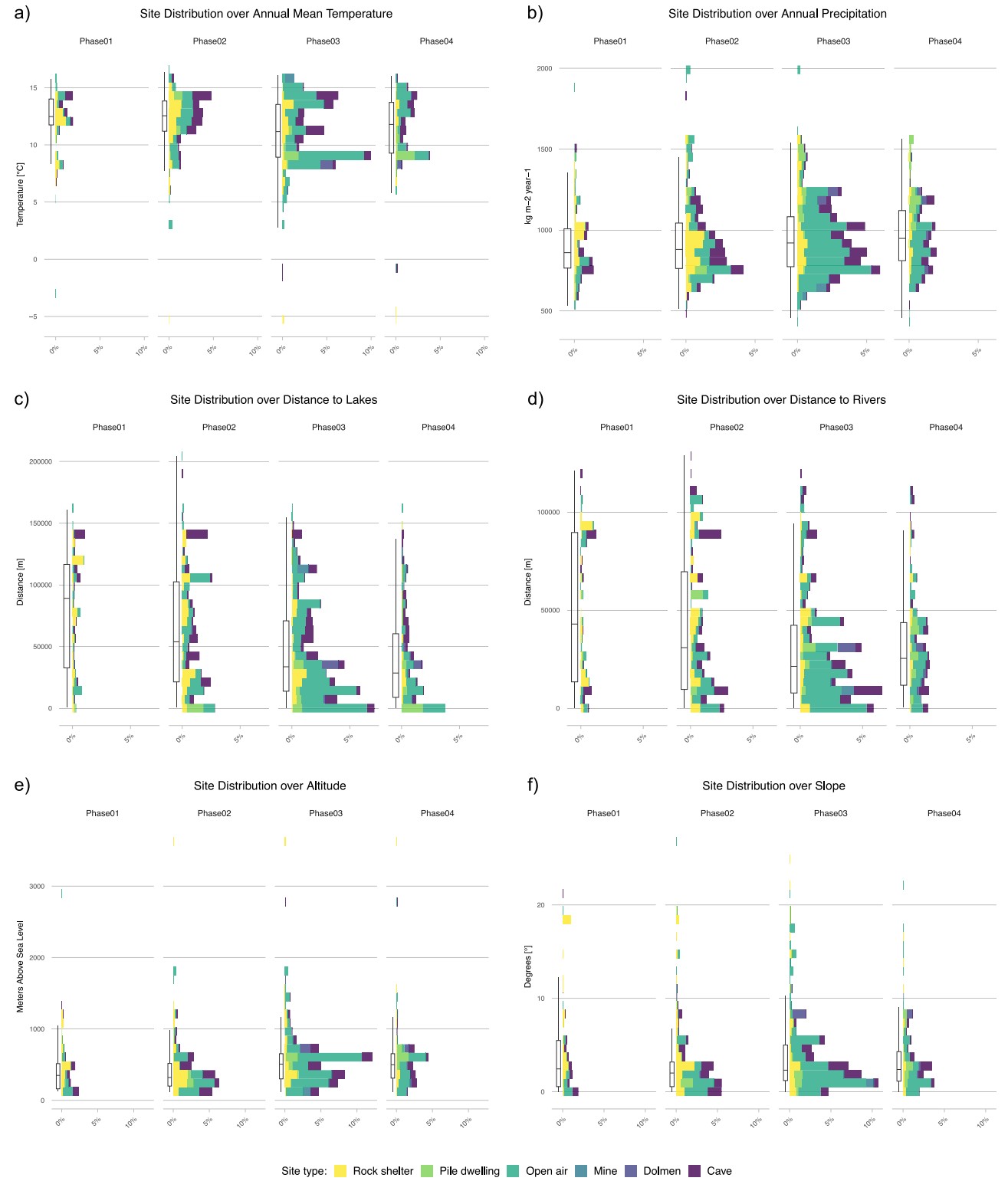

Fig. 3 | Distribution of Site types over paleoclimate and environmental variables per Phase. Each panel shows the overall distribution (box plot) of all sites and the distribution of site types (histograms) over Annual Mean Temperature (a), Annual Precipitation (b), Distance to lakes (c), Distance to rivers (d), Altitude (e) and Slope (f) per Phase. The histograms display 30 equally wide bins showing the percentage distribution of site types over the variables. The boxplots show medians, first and third quartiles (hinges), and minimum and maximum values no further than 1.5*IQR from the hinge where IQR is the inter-quartile range (whiskers). Details about sample numbers are provided in Supplementary Table 2. Source data can be found at https://doi.org/10.5281/zenodo.14253277.

similar interquartile ranges (boxes) with slightly higher median values during Phase 01 and 02. A similar altitude shift between Phase 02 and Phase 03 is observed when the median altitude of the most suitable areas increases from 304 m.a.s.l. to 360 m.a.s.l and their distribution spreads over a larger range (see Fig. 6c, d). Indeed, it was around 4500 BC that we could also observe a shift in the type of preferred landscapes. While during the first two phases, the most suitable areas were mainly located at the lowest altitude, with warmer temperatures and

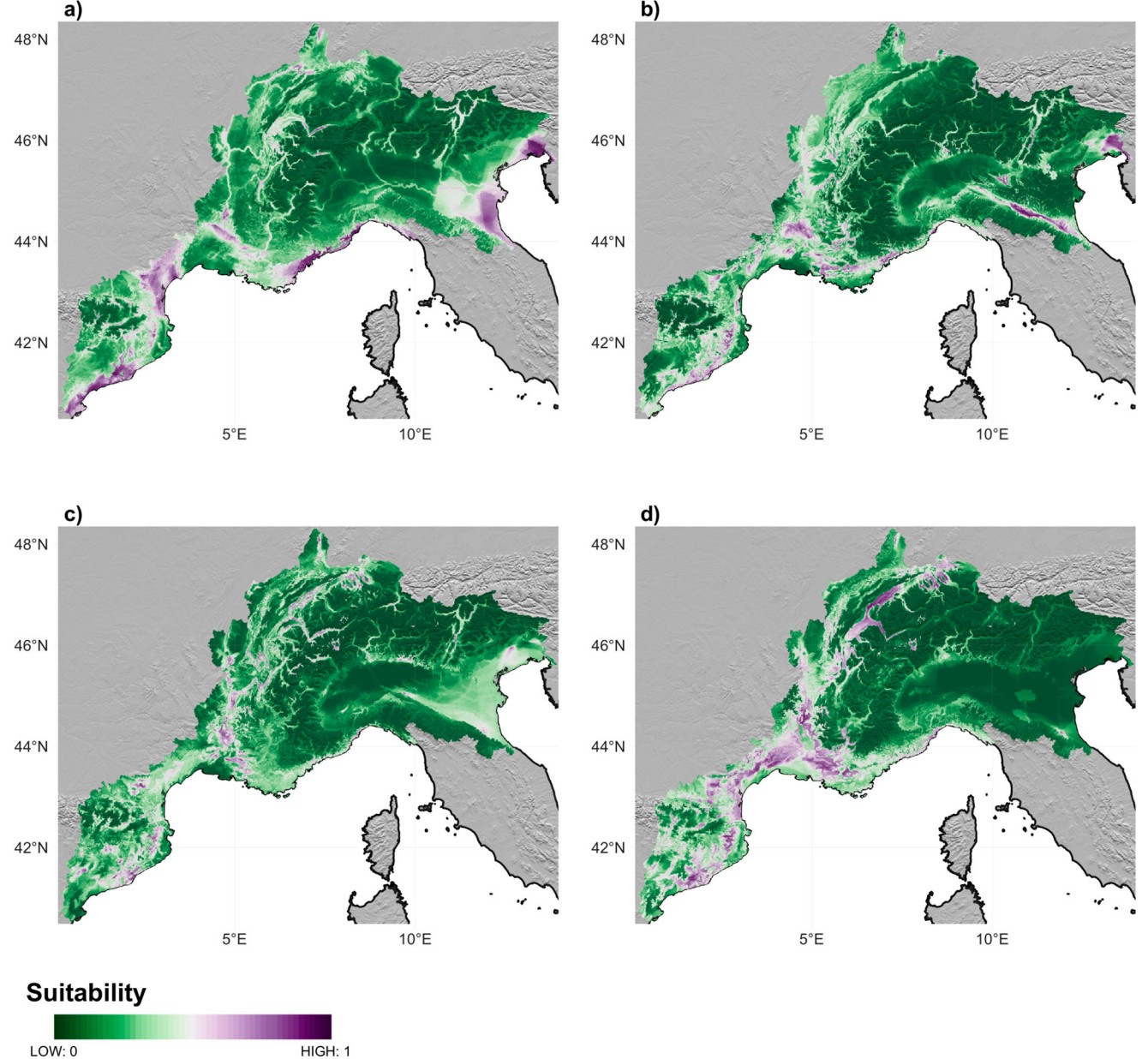

**Fig. 4 | Suitability maps resulting from the Random Forest models for the four Phases.** Phase 01 is shown in panel (**a**), Phase 02 in panel (**b**), Phase 03 in panel (**c**) and Phase 04 in panel (**d**). High suitability areas are coloured in purple, while low suitability areas are coloured in green. Source data can be found at https://doi.org/10.5281/zenodo.14253277.

low precipitation values, in time they shifted towards higher elevations, in colder and more humid environments. (for a complete overview of the distribution of the predicted areas over all variables see Supplementary Fig. 7).

**Agricultural variability and Crop niches**

We also analysed the ecological niche of each of the main crops over the entire period. These might differ individually from the general trends shown by the sites since not all the crops were present at all sites during all phases (see Fig. 7a).

Figure 8 reproduces the distribution of crop occurrences over Annual Mean Temperature and Mean Annual Precipitation. It shows that for the entire period analysed, naked wheat clustered around higher temperatures, between 12 °C and 15 °C, and annual precipitation values of ca. 800 mm per year (see also Supplementary Fig. 8a, k), spreading in warmer and more arid environments. Glume

wheat (einkorn, emmer and Timopheev's wheat), similar to naked wheat, cluster with preference in areas with relatively high Annual Mean Temperatures (ca. 14 °C to 15 °C) and mean annual precipitation around 800–880 mm per year. Poppy and apple/pear instead are found in more humid and cooler environments, with precipitation values around 1000–1250 mm and temperature values of less than 10 °C. Pea and hazel, and to a lesser extent lentils and oak, are found both in warmer and drier as well as in humid and cooler conditions.

The results (Fig. 9) show that mild winters with lower precipitation values were suitable for barley, glume wheat, naked wheat, and oak (clusters around temperature values of the coldest quarter above 5 °C and precipitation of the coldest quarter around 200–250 mm, see also Supplementary Fig. 8j,r). In places with harsher winters (temperature values of the coldest quarter below 0 °C and precipitation of the coldest quarter above 250 mm), wild crops such as apple/pear and

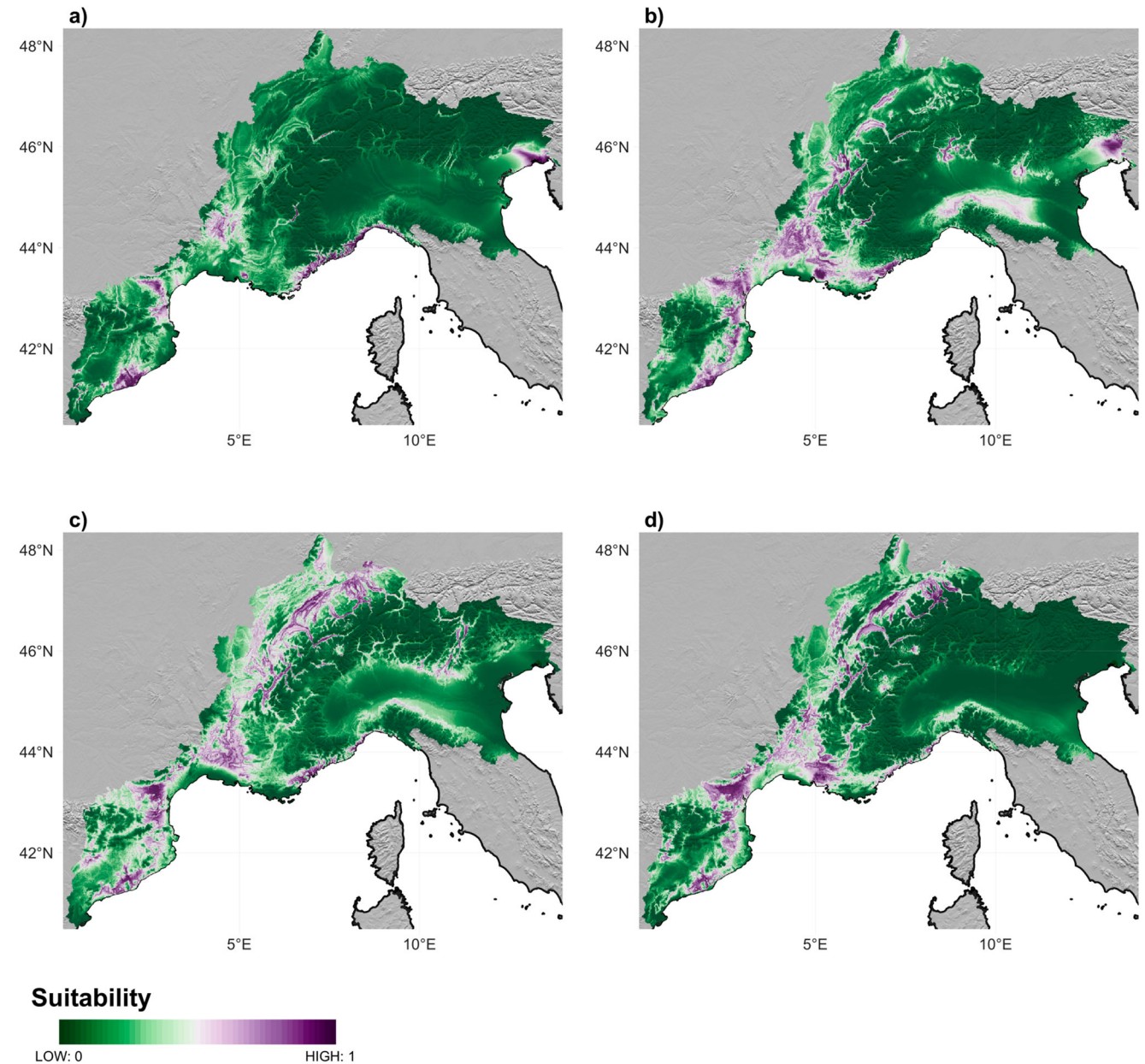

**Fig. 5 | Suitability maps resulting from the MaxEnt models for the four Phases.** Phase 01 is shown in panel (**a**), Phase 02 in panel (**b**), Phase 03 in panel (**c**) and Phase 04 in panel (**d**). High suitability areas are coloured in purple, while low suitability areas are coloured in green. Source data can be found at https://doi.org/10.5281/zenodo.14253277.

hazel were complemented with oil seeds (flax and opium poppy). Lentils and peas are found in both types of environments.

## Discussion

The main objective of this research was to identify changes in human and crop niches and their interrelationship with climatic changes across the Western Mediterranean area, based on a big database and a data-driven approach using machine learning techniques. While computational and quantitative modelling techniques offer an exciting new opportunity to create and empirically test more explicit models of past human ecological dynamics and agriculture development strategies, one must keep in mind that these approaches carry also some limitations. Accessing and manipulating large datasets across different sources requires, on the one side, significant computational power and processing time and on the other side, complementary and multi-disciplinary expert knowledge. Moreover, the inherent bias within the

archaeological and archaeobotanical datasets represents today a challenge for many modelling exercises, in particular here where the available site and crop samples with associated C14 dates are likely under-representative for some of the regions included in the study area (e.g., the Catalan coast or the regions of northern Italy and the Alps), and for some of the periods identified (Phase 04 in particular). The state of archaeobotanical research may indeed influence the suitability models of some of the crops. This is the case of the opium poppy, for instance, a crop of Mediterranean origin that spread to central Europe together with other Neolithic elements (and possibly populations) most likely from the Western Mediterranean area[43,44]. Our analyses could suggest its great suitability to wet and cold climates, but actually, they show how quickly the plant was able to adapt to and thrive in new climatic conditions under cultivation. The spatial and temporal bias may be due to preservation issues due to a different distribution of research efforts on the territories or to the fact that

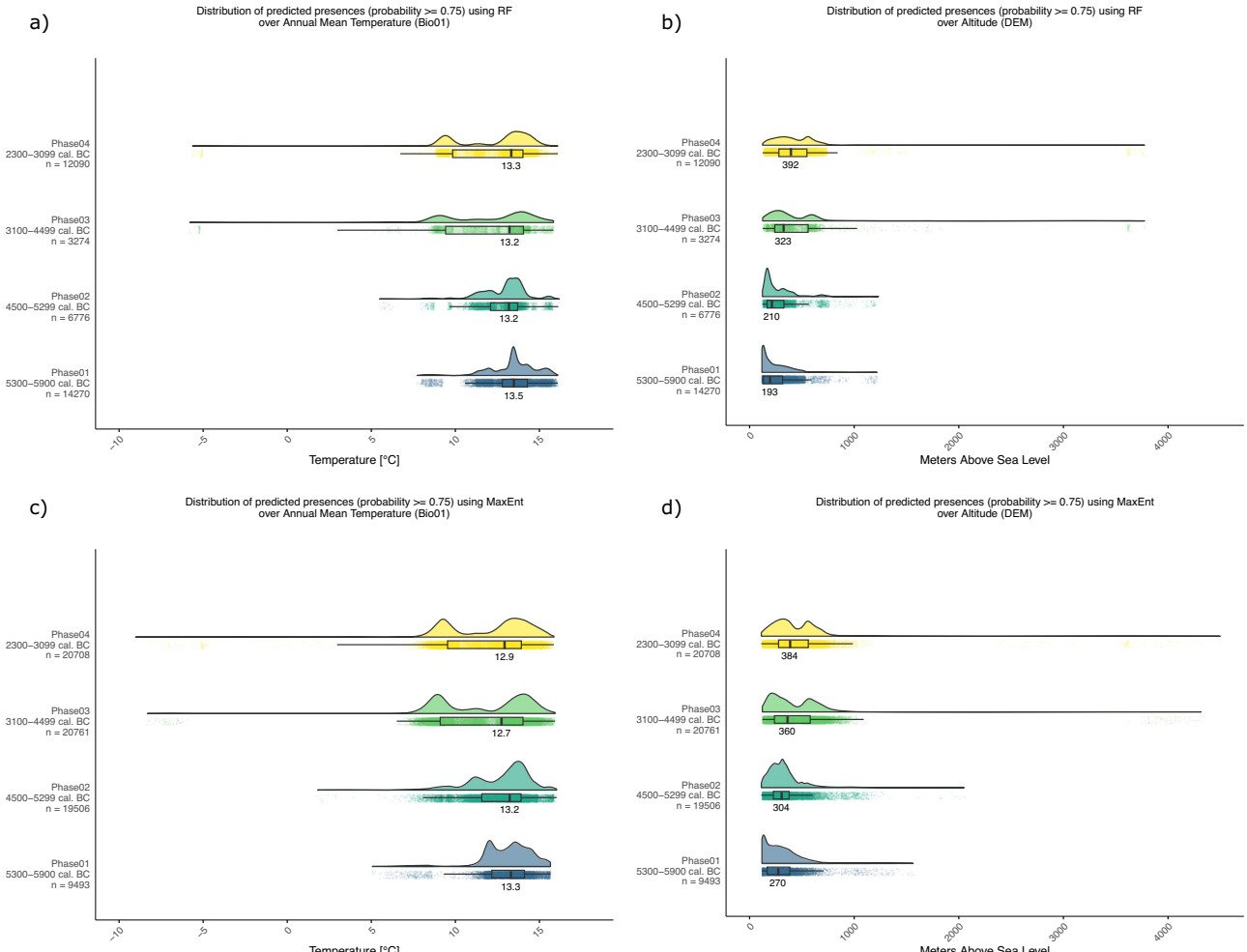

**Fig. 6 | Comparison of the distribution of predicted high suitability areas using both models over Annual Mean Temperature and Elevation. a** shows the distribution of predicted high suitability areas (≥ 0.75) modelled using Random Forest (RF) over Annual Mean Temperature (Bio01). **b** shows the distribution of predicted high suitability areas (≥ 0.75) modelled using Random Forest (RF) over Elevation (DEM). **c** shows the distribution of predicted high suitability areas (≥ 0.75) modelled using Maximum Entropy (MaxEnt) over Annual Mean Temperature

(Bio01). **d** shows the distribution of predicted high suitability areas (≥ 0.75) modelled using Maximum Entropy (MaxEnt) over Elevation (DEM). Ridge lines on each panel show kernel density estimates, and the boxplots show medians, first and third quartiles (hinges), minimum and maximum values no further than 1.5*IQR from the hinge where IQR is the inter-quartile range (whiskers). Points show the jittered distribution of data points. Source data can be found at https://doi.org/10.5281/zenodo.14253277.

archaeological sites have a greater chance of being discovered if they are repeatedly used in time. Lastly, our study can also be partly limited by the absence of soil-related variables used in the modelling procedure, such as indices of soil types, texture, and vegetation distribution[55]. All of these drawbacks need to be considered in the discussion of the results generated.

In this study, we applied both spatial statistics analysis and machine learning/species distribution modelling comparison techniques using RF and MaxEnt. One can consider these two algorithms as complementary methods. RF is an ensemble learning method based on decision trees, constructing multiple decision trees during training that outputs the average prediction of the individual trees. MaxEnt, however, is a probabilistic modelling approach that aims to find the distribution of maximum entropy given a set of constraints. RF tends to capture complex interactions between variables, while MaxEnt might provide more generalised response curves. We believe that comparing these two techniques allowed us to take advantage of the strength of the two models and to be more confident in the comparable predictions.

The combined analyses performed in this paper prove very useful to answer questions regarding agricultural decision-making and spreading in the past. Phase 01 presents a relatively narrow niche. The

strong seasonality and the colder climatic conditions favoured glume wheat as a predominant taxa, possibly due to their resilience capacity, within broadly diverse crop assemblages. These first two Phases, Phase 01 and Phase 02 also witness a cluster of settlements suitability areas mainly distributed along the Mediterranean shores. In Phase 03, a warmer and more humid phase allowed farmers to migrate to drier internal zones and to colder temperatures, where they mostly took the set of crops from the previous phase. The optimal climate conditions during this Phase, along with the ease of processing of these specific taxa prior to consumption might have led the farmer communities to prefer these among others[56]. Several factors might have influenced this decision, such as a higher population density or a possible population boom, as observed in several archaeological proxies from the area[57] (and in Europe)[58]. So not only could these cereals have been suited for the climate, but also the most productive and interesting to sustain a growing population inhabiting a broader niche. With the general increase in temperature and decrease in seasonality (see Fig. 2b), the climatic niche for barley, naked and glume wheat shifted inland towards higher altitudes over time. This might have allowed farmer communities to expand and occupy new territories. Similar observations were made in the natural environment of the alpine and

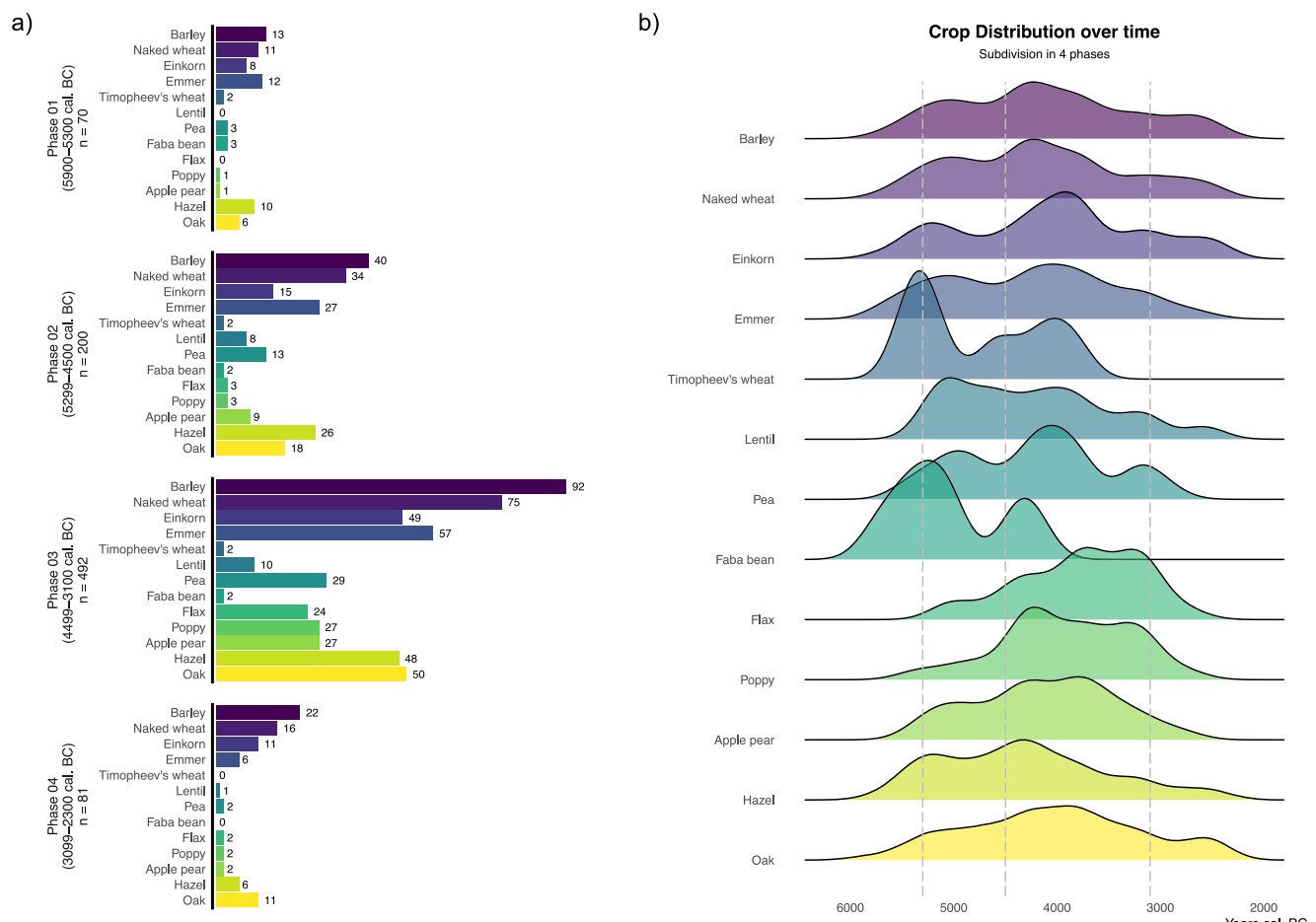

**Fig. 7 | Quantified crop occurrences per phase (a) and density of crop occurrences over time (b). a** The bars show the number of occurrences in the database for each type of crop per chronological phase. **b** Coloured areas show Kernel Density estimates of crop occurrences over the entire period for each crop. The vertical dashed lines indicate the limits of our 4 chrono-phases. Details about sample numbers are provided in Supplementary Table 3. Source data can be found at https://doi.org/10.5281/zenodo.14253277.

subalpine areas of the central Pyrenees during the first half of the Holocene period[20]. Some authors had interpreted these sites at higher altitudes as evidence of increased site specialization[21], but our results would support an expansion into the highlands of year-round farming sites during a period of climatic amelioration, which would have a more or less similar economy and would be tightly networked in comparison to previous phases. The networks during this period have been extensively analysed using many different types of artifacts (i.e., pottery[59,60]). Our results suggest that farmers responded to climate change by moving to higher altitudes (broadening their niche) and not as a shift to specialization in particular products, but rather to maintain the resilience of the whole mixed farming network. During Phase 04, wetter conditions prompted farmers to adapt and favour wet-tolerant crops. In colder and wetter environments, wild crops such as oil seeds flax, opium poppy, pea or apple/pear might have been more regularly added to the communities' diets in order to compensate for the difficulty of growing naked wheats as in earlier periods. Emmer seems to partially replace free-threshing wheat in wetter and colder areas.

The results highlight in particular three relevant moments of change at a large scale and of high relevance: (1) A first one in correspondence of 5300 BC approx., when the climate became drier in the NW Mediterranean area, and thus the early farmers had the possibility to choose the most appropriate crops to grow and keep growing at specific locations. Until this moment, glume wheat was the predominant taxa within broadly diverse crop assemblages and the settlements clustered mainly along the Mediterranean shores, but during

Phase 02, naked wheat and (naked) barley become more important, and glume wheat become residual (as also observed in other works[7]); (2) This situation started to change around 4000 BC, Specifically, the peak of naked wheat occurrences (Fig. 7b) happens at a moment when Mean Annual Temperature and Precipitation are at their highest (in Phase 03). After this moment, glume wheats, particularly einkorn, gain importance. It is possible also that not only climate variations but also the emergence of main storage pests, such as the grain weevil, might have played a role in the clear shift of the crop spectrum, particularly visible in the Mediterranean regions, as recently suggested by other works[47,56]. The spread of these pests could have been facilitated by the active networks that functioned at the time. (3) Finally, a last moment of major change can be identified around 3100 BC, when the wetter climate in the alpine Foreland prompted the farmers to adapt to the new conditions and to make different choices in order to face the new climatic/environmental conditions. For example, we see that oil seeds like flax, opium poppy, pea or apple/pear, have been found in colder and wetter environments than barley, glume wheats and naked wheats. These wild crops might have been more regularly added to the communities' diets in order to compensate for the difficulty of growing naked wheat in such environments (see also Steiner et al.[61]). Changes towards more widespread cultivation of einkorn and emmer have also been pointed out for regions such as Southern France[62] and Switzerland[35]. Our analyses indicate that these changes could have been, to some extent, an adaptation to new climatic conditions, although social factors cannot be excluded.

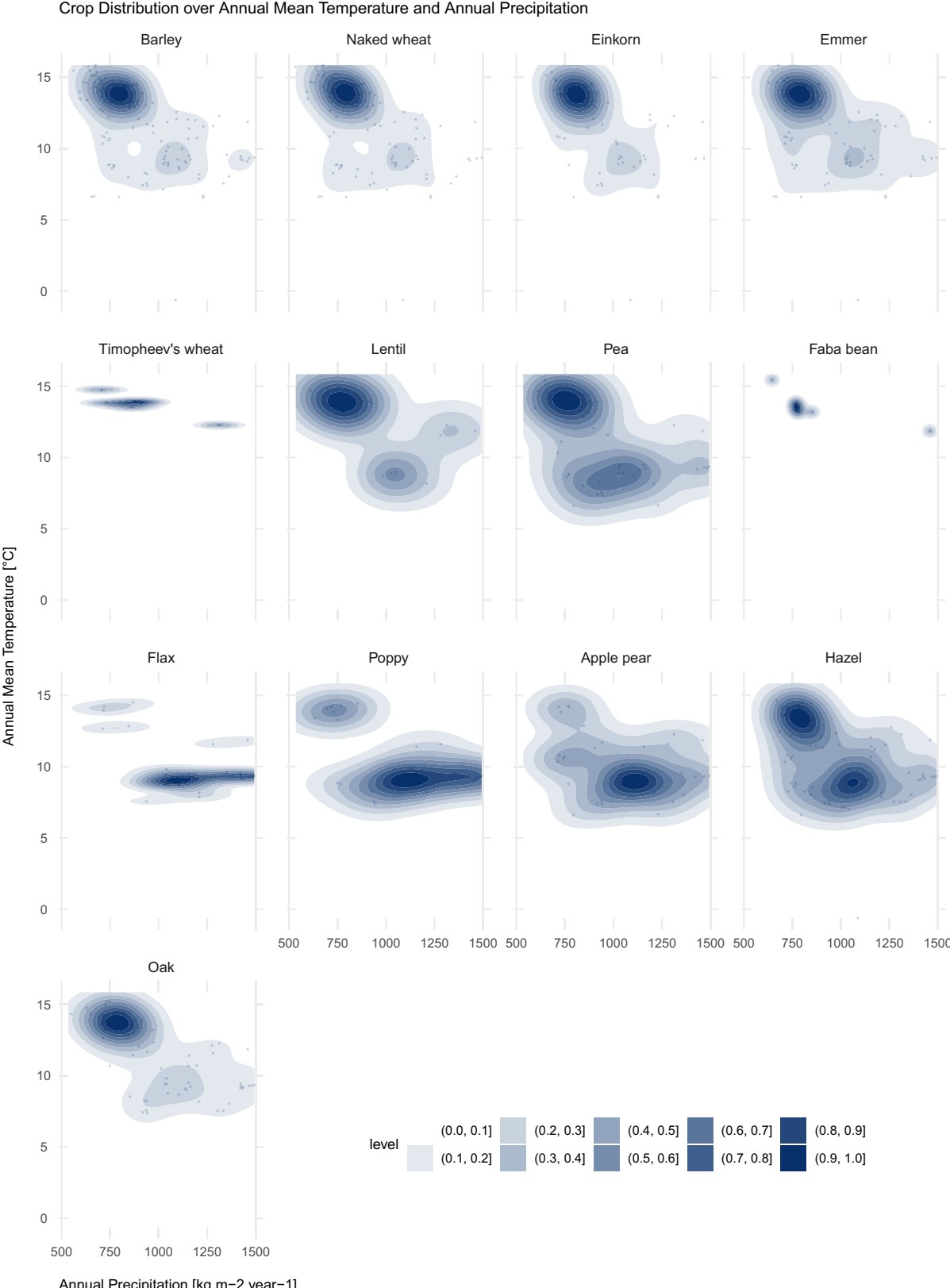

**Fig. 8 | Distribution of crop occurrences over Annual Mean Temperature and Annual Precipitation.** Blue areas in each panel show 2D kernel density estimates of the distribution of crop occurrences over Annual Mean Temperature (*y*-axis) and Annual Precipitation (*x*-axis) over the study period. Darker blue indicates higher density and light blue indicates low density. Details about sample numbers are provided in Supplementary Table 3. Source data can be found at https://doi.org/10.5281/zenodo.14253277.

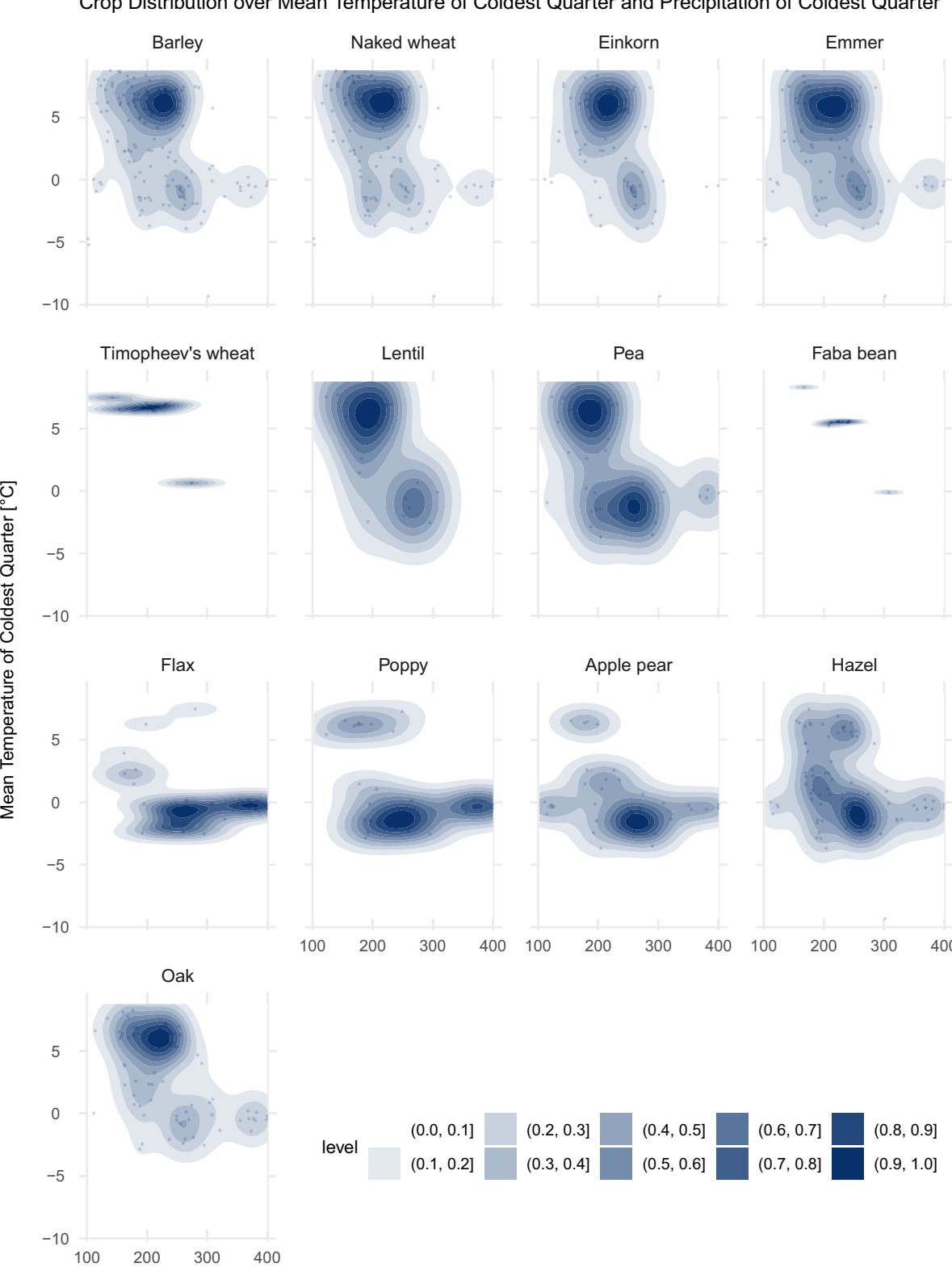

**Fig. 9 | Distribution of crop occurrences over Mean Temperature and Precipitation of Coldest Quarter.** Blue areas in each panel show 2D kernel density estimates of the distribution of crop occurrences over Annual Mean Temperature (*y*-axis) and Annual Precipitation (*x*-axis) over the study period. Darker blue indicates higher density and light blue indicates low density. Details about sample numbers are provided in Supplementary Table 3. Source data can be found at https://doi.org/10.5281/zenodo.14253277.

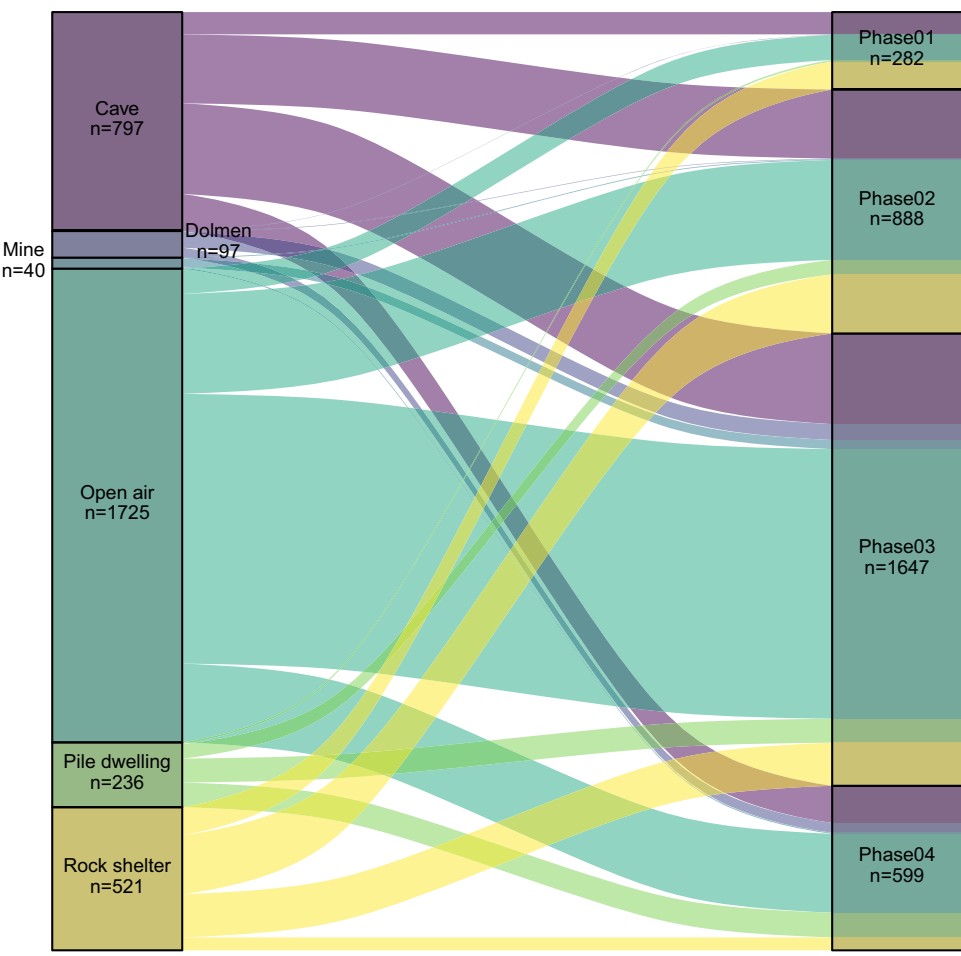

**Fig. 10 | Distribution of sites per type and phase.** The Alluvial diagram shows the distribution of sites per site type (left) and per phase (right). Source data can be found at https://doi.org/10.5281/zenodo.14253277.

Comparably to recent works that made use of a similar methodology, for more specific ecological analyses[63,64], we have identified correlations between temperature and precipitation increases/decreases and the overall settlement distribution patterns and crop diversification for the period 5900-2300 BC cal. The results we obtained are consistent with previous studies[64,65] that highlighted considerable challenges related to climate and climatic changes being among the key drivers of Neolithic human dynamics and agricultural adaptation and resilience strategies[30,66], which furthermore suggest how small-scale communities could have developed different adaptive or resilient strategies to face climatic-based limitations in a given time and space[67].

We conclude that human niche changes and crop changes in the Neolithic period of the region are independent of each other. Crop niches prove to have a plasticity beyond initial expectations from present-day crops and human niche breadth expansion is possible without crop changes. Crop changes resulting from short-term and long-term climatic oscillations were instead detected. As archae-obotanical and climate datasets gain higher resolution, paleo-climate and paleo-environmental modelling approaches become the road to be travelled for the future to reconstruct ancient crop ecologies and, hence, to refine uniformitarian inferences.

## Methods

### Archaeological and archaeobotanical data
A database consisting of a total of 3416 geo-referenced archae-ological sites, with associated radiocarbon dates (calibrated using

OxCal v. 4.4.2 and the atmospheric curve IntCal20) falling within the timeframe considered in this research was initially drawn from *AgriChange_14Cdatabase*[49]. This open-access dataset contains a unique collection of inventoried sites located between the Upper Rhine, the Po, the Rhone and the Ebro valleys, obtained from a combination of own fieldwork and the published literature. It includes six different types of archaeological sites (we grouped *Chasm* type in the *Rock shelter* category), among which *Open air* sites with multiple occupation phases account for over half of the occurrences, followed by *Cave sites*, *Rock shelters* and *Pile-dwellings* (see Fig. 10 and Supplementary Table 2). The chronological phases used in this paper follow previous work[49,68,69] and correspond to the main socioeconomic dynamics in the region.

The fields extracted from this database and retained for the present research concern the specificity of the sites (ID, name and type of the site, their geographic coordinates (GCS WGS 1984 using decimal degrees as angular unit)) and the dating information (the corresponding calibrated *mean*). When sites were occupied multiple times, and samples lay in different stratigraphic units, these were treated as individual sites with their respective environmental conditions.

In addition, a second database[33,70], collecting archaeobotanical information for 843 archaeological sites, was integrated and processed. Thus, the archaeobotanical dataset assembled for this study provides a synthesis of the recorded presence of taxa (only charred remains were considered) at a site level for the region of interest, with a specific reference to the seeds and the crop typologies described (the term "crop typology" refers here to economically important,

potentially managed, plants and not only to traditional crops). We merged this information in a unique database, and hence, all available crop records were georeferenced and attributed to a spatiotemporal dimension (Fig. 7 and Supplementary Table 3).

## Paleoclimate variables

The selection of paleoclimate records was essentially based on a set of criteria, including high dating reliability, high time resolution and a spatial extent that could cover the entire region in the exam. Specifically, we use a set of 18 reconstructed climatic variables (see Supplementary Table 4) from the mid-to-late Holocene for land surface areas, selected based on their relevance and interpretability for our research. These variables are retrieved from the open access dataset *CHELSA-TraCE21k v.1.0*, downscaled[51,71] and derived from the CCSM3_TraCE21k model to a 30 arcsec resolution using the CHELSA V1.2 algorithm[71], which covers time steps of 100 years for the last 21'000 years, with minimum and maximum temperatures, surface precipitation, and paleo-orography information. It contributes to creating a paleoclimatic envelope that best matches the spatiotemporal distribution of our settlements and crops. The data were read as GeoTiff using the *raster* package[72] in *R*[73]

## Environmental variables

In addition to the paleoclimate variables (see Supplementary Table 4) were computed and used to describe landscape characteristics, such as the digital elevation model (DEM) expressing the elevation and derived from the same *CHELSA-TraCE21k v.1.0* dataset. Although the elevation may not be an optimal proxy for describing the landscape, as it may introduce bias in the modelling procedure and catalyse predicted areas[74], we nevertheless decided to include it here to evaluate if adaptation to height might have played a role in the distribution of the early Neolithic farmers and their specific subsistence activities as done in other works[75]. We further derived the slope, which defines the steepness of the terrain, and the terrain ruggedness index (TRI), which is the mean of the absolute differences between the value of a cell and the value of its eight surrounding cells, using the *terrain* function in the *terra* package[76], and calculated the proximity to important water resources[77] as reconstructed permanent lakes and rivers in a GIS environment (see the tool *Euclidean distance* in ArcGIS 10.8 – ESRI).

## Habitat Suitability and Niche models construction

The methodology developed in this study is primarily based on machine learning techniques and on species distribution modelling (SDM) approaches, linking known species localities with predictor variables to assess patterns of species occurrence and habitat suitability[78,79]. Such methods, well established in ecology and biogeography[80], evolution and more recently in conservation biology and climate change research[78,81–83], have only recently seen first applications in archaeology and, more specifically in research related to archaeobotanical studies, as modelling tools for analysing occurrence data and for predicting human habitat suitability[29,66]. Especially, extensive literature supports the use of advanced quantitative and machine learning methods to reconstruct empirical ecological settings of ancient human populations and their subsistence activities[25,84–86], as well as to examine potential environmental and climatic changes and their ecological consequences in modern and future scenarios.

Although there is a variety of algorithms with different levels of complexity for SDMs[87], only a limited number of algorithms are being applied in archaeology (often Maximum Entropy alone) and thus we decided to test and compare the algorithms of *Random Forest* (RF)[88] and *Maximum Entropy* (MaxEnt)[89] and to explore and discuss their results, as also suggested by several authors[80,90].

RF is an ensemble decision tree method of ML based on classification and regression tree algorithms. It is widely recognized for its capacity to produce good predictive models with few sparse data[91]. Further, it can tolerate noise overfitting and can handle a large number of predictors and their interactions[92–95], although, it is still in its infancy in archaeological studies, with few applications to different research branches[96–98]. This technique has been applied here to investigate the relationships between a categorical dependent variable (Neolithic settlements) to a set of predisposing factors such as ecological and climatic variables (topography, paleoclimatic data, etc.), to identify if and to what extent the ecological setting of the study area acts as a predictor in the determination of the dependent variable (settlements) and by this means contributes to the shaping of an eco-cultural niche.

MaxEnt builds upon the principle of maximum entropy and is used to approximate a target probability distribution of data occurrences that is closest to uniform and subject to environmental/climate constraints[99]. Being a generative approach rather than discriminative, it shows several advantages when the amount of training data is limited and when only presence/occurrences data is available[100]. For its high-performance rate, MaxEnt is mainly preferred in SDMs in the field of ecology and palaeoecology and has repeatedly shown to be an invaluable tool in a wide number of applications[101–103], but yet only a few and very recent applications of this specific algorithm can be enumerated in archaeology[75,104,105]. Both algorithms can utilize continuous and categorical data and can incorporate interactions between different variables. They are considered among the leading data-mining ML methods for their high accuracy prediction[93].

Our modelling pipeline conceptually follows the research workflow defined in the most recent literature[65,100,106], as well as by research exercises for modelling population distribution, agriculture and crop niches on different spatiotemporal scales[25,66].

Data preparation was performed both in *ArcGIS 10.8, ArcGISPro* and in *R*. The model calibration, final model computations, post-modelling analyses, and assessment were performed in *R*.

We used 18 downscaled Bio (paleoclimate) variables extracted for the study area and limited to the period 5900-2300 BC to spot climate trends for the specific region and period[29]. We then combined this climatic information, along with the computed five environmental variables (elevation, slope, TRI, distance to main rivers, distance to main lakes) to classify the local ecological envelope for each site type and location.

The paleoclimatic and environmental variables retained as model inputs in the subsequent modelling procedure were further used to build two HS Models (based on the *randomForest*[107] and *maxnet*[108] packages available at *cran.r-project.org/web/packages/randomForest/* and *cran.r-project.org/package = maxnet*).

For each of the four phases, we imported the occurrence data (archaeological sites) and the predictor variables (paleoclimate and environmental), defining the spatial resolution and extent of the analysis to a 30-arcsecond cell size.

To avoid spatial autocorrelation affecting the predictors and the model results[18] and to reduce the dimensionality of the initial pool of 18 paleoclimates and five environmental variables, we used the non-parametric Spearman correlation coefficient test for each variable set (paleoclimatic and environmental). From the list of the paleoclimate and environmental variables (as shown in Supplementary Table 4), we retained those predictors with correlation values of $r < 0.25$. Where several variables form highly correlated clusters ($r > 0.75$), we selected the one predictor from each cluster least correlated to all other variables. Slope and elevation are correlated by less than 0.01 more than the threshold of 0.75. Nonetheless, as the dataset contains high altitude sites, we decided to keep both variables, as we considered them to be important factors to detect relations between site locations and their environmental surroundings, at play here both in past human mobility choices and adaptations strategies, thus retaining 9 paleoclimate and 4 environmental variables as input in the modelling process. These include Mean Diurnal Range (Bio02), Temperature

Seasonality (Bio04), Mean Temperature of Warmest Month (Bio05), Mean Temperature of Wettest Quarter (Bio08), Mean Temperature of Driest Quarter (Bio09), Precipitation of Wettest Month (Bio13), Precipitation Seasonality (Bio15), Precipitation of Driest Quarter (Bio17) and Precipitation of Coldest Quarter (Bio19), as well as Distance from Lakes (D_Lakes), Distance from Rivers (Dist_Riv), Slope (Slope) and Altitude (DEM) (see Supplementary Fig. 9).

We performed both model fittings with spatial split-block cross-validation techniques. The data are split into k-independent subsets, and for each subset, the model is trained with k-1 subsets and evaluated on the $k^{th}$ subset[109]. The optimal block size was selected based on the spatial autocorrelation structure of the predictors using the *spatialAutoRange* function in the *R* package *blockCV*[110] and was set at 170 km X 170 km.

We ran the RF model in classification mode, and because our dataset does not contain confirmed absences, we generated a set of random pseudo-absences over the landscape, equal in number to the occurrence dataset. Bootstrapped comparisons were repeated 1000 times to ensure that sites and their surroundings were treated as a whole, and to generate a distribution of AUC values using different training data sets, while four predictors were chosen at each split.

Unlike RF, MaxEnt uses presence-only data to predict the distribution of species based on the theory of maximum entropy. We run it with presences only and a group of 10,000 random background points, as this amount of background data is considered large enough to provide an accurate representation of the study area, while a larger background sample increases computation time without improving modelling performance[111,112]. The model prediction function was set on *cloglog* transform[108], it is the most appropriate method for estimating probability of presence and since it gives a better result over logistic when bias correction is used[108]. Other settings were left at their default values.

Eventually, we obtained eight suitability maps, one for each Phase, in which each raster cell was assigned with the relative index of occurrence of the archaeological settlement (ranging from 0 for predicted absence to 1 for predicted presence – low to high suitability). We further generated variable importance rankings, partial dependence and response plots to reveal the importance of each predictor variable in the prediction of site occurrence.

### Model validation and performance assessment

In this study and in spatial data analysis more broadly, test dataset observations are often situated near training dataset observations. This spatial proximity can cause an overestimation of model performance due to spatial autocorrelation, where nearby observations share similar attributes.[92] To address this issue, training and testing datasets should be geographically separated. Spatial k-fold cross-validation is a common method to achieve this. The dataset is divided into k non-overlapping groups, the model is trained on k-1 groups, and performance is evaluated on the remaining group. This process is repeated k times, and the error estimates are averaged to provide an overall performance metric. For this study, we employed 5-fold spatial cross-validation, implemented with the blockCV package[109].

Model performance for both algorithms was assessed on test data as the *Area Under the Receiver-operating characteristic curve* (AUC), which represents the capacity of the models to distinguish between the presence and absence or background points. This measure is independent of threshold selection, making it a powerful tool for assessing model performance[113]. The accuracy of the two models was assessed through the AUC curves, which are presented in the Supplementary Fig. 10. This graphical method illustrates the relationship between the true positive rate (TPR) and the false positive rate (FPR), both represented as percentages of the total cases. True positives (TP) and true negatives (TN) refer to correctly identified outcomes, while false positives (FP) occur when a prediction incorrectly labels an outcome as a presence when it is actually an absence or background point.

Conversely, false negatives (FN) happen when an outcome is wrongly classified as an absence or background point instead of a presence. AUC can generally range from 0 to 1, where a score of 1 indicates perfect discrimination and values < 0.5 indicate a predictive performance worse than random. AUC values for RF models range between 0.64 and 0.78 accuracy on the testing dataset, while the AUC values for MaxEnt models range between 0.69 and 0.86 on the testing dataset, indicating that RF model predictions are slightly more accurate than MaxEnt for Phase 02 and 03 and MaxEnt model predictions are more accurate for Phase 01 and 04.

### Statistics

R 4.1.2 version was used for all statistical and ML modelling analyses. Parameters such as sample size, number of replicates are reported in the Text, Figures and Figure Legends.

### Reporting summary

Further information on research design is available in the Nature Portfolio Reporting Summary linked to this article.

## Data availability

The source archaeological data used in this study are available at: https://doi.org/10.5334/joad.72. The paleoclimatic dataset used in this study was published by: Karger, D. N., Nobis, M. P., Normand, S., Graham, C. H., Zimmermann, N. E. (2020). CHELSA-TraCE21k: Downscaled transient temperature and precipitation data since the last glacial maximum. EnviDat. https://doi.org/10.16904/envidat.211 and is available at: https://envicloud.wsl.ch/#/?bucket=https%3A%2F%2Fos.zhdk.cloud.switch.ch%2Fchelsav1%2F&prefix=chelsa_trace%2F. The computed analyses, archaeological and archaeobotanical (crop) datasets produced in this study are available on the corresponding author's GitHub at https://github.com/MaCasti21/Nat-Comm_Castiello_2024. Data to generate all figures can be found on the following Zenodo repository: Castiello, M.E., et al., Understanding the spread of agriculture in the Western Mediterranean (6th-3rd millennia BC) with Machine Learning tools, Nat-Comm_Castiello_2024, https://doi.org/10.5281/zenodo.14253277, 2024.

## Code availability

Code to reproduce the results and generate all figures can be found on the following Zenodo repository: Castiello, M.E., et al., Understanding the spread of agriculture in the Western Mediterranean (6th-3rd millennia BC) with Machine Learning tools, Nat-Comm_Castiello_2024, https://doi.org/10.5281/zenodo.14253277, 2024 and on the corresponding author's GitHub at https://github.com/MaCasti21/Nat-Comm_Castiello_2024.

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

## Acknowledgements

The study was funded by the GroundCheck Research Cluster of the German Archaeological Institute and the Schweizerischer Nationalfonds zur Förderung der wissenschaftlichen Forschung (Swiss National Science Foundation) in the framework of the SNSF professorship of Ferran Antolín (Grant Number PP00P1 170515). We thank Dr. Marj Tonini, Dr. Alejandra Moràn Ordonez, and Dr. Andreas Angourakis for the insightful exchanges.

## Author contributions

M.E.C.: Conceptualisation; Methodology; Formal analysis; Visualisation; Writing original draft; Manuscript Review and Editing. E.R.: Climatic analyses: Conceptualisation and Methodology of the climate analysis; Review original draft. H.M.G., A.J., G.P. and F.A.: Archaeological and archaeobotanical data Collection and Processing. F.A.: Obtained funding; Supervised the study; Writing original draft; Manuscript Review and Editing. All authors reviewed the manuscript and provided feedback.

## Competing interests

The authors declare no competing interests.
