## [Peer Review file · Nature Communications]

Understanding the spread of agriculture in the Western Mediterranean (6th-3rd millennia BC) with Machine Learning tools

Corresponding Author: Dr Maria Elena Castiello

Version 0:

Reviewer comments:

Reviewer #1

(Remarks to the Author)

The project described in this paper is interesting and ambitious, and appears to have been carried out with admirable thoughtfulness and attention to detail. I enjoyed learning something about the work, and see this kind of combination of data integration and modeling as an important and promising route forward for archaeology. However, I think that this paper needs substantial revision in order to be publishable. In its current form, I find the modeling process hard to follow, making the conclusions difficult to evaluate – and the conclusions themselves seem more focused on the successes in creating a model than on what can be learned from it.

On p25 the authors summarize the goal of the project: “the development of a quantitative data driven approach to model and approximate the features of the ecological niches exploited by the first Neolithic farmers.”

Conceptually this begins with locational modeling, i.e. characterizing the regions in which archaeological sites are found. Taking paleoclimate data into account provides a means of taking into account that the characteristics of these locations may have differed when they were selected as settlement locations. I see this as sound practice, and have done similar things myself. I might quibble over methodological details (the authors don't seem to have taken into account issues of settlement persistence / path dependence, interestingly explored in e.g. Verhagen et al. 2016), but in fact I think that the exploration of RF and MaxEnt is a step forward in most respects. My concerns don't have to do with the modeling per se but rather the way in which it is presented and the conclusions drawn from it.

Presentation

The most succinct statement of the paper's import does not come until p4 (Lns99-101), and then it is a statement of ambition rather than of findings:

“This paper aims at exploring the impact of environmental and climatic constraints on the distribution of the first farming populations in the Western Mediterranean area during the period of 5900-2300 cal. BC.”

This characteristic is repeated throughout the introduction: the paper is framed more as a grant proposal (“this is what we will attempt to do”) than a report of results (“this is what we found when we did X”). On Lns 117-118, for example (“In order to investigate such hypotheses, we developed an innovative multi-proxy and interdisciplinary approach”) and in the concluding lines (127-133) of the Introduction (where the verbs employed are “explore and reveal”, “examine”, “evaluate”, “map”, “assess”, “diagnose”). Of course doing these things is fundamental to scientific investigation, but for the purposes of the high-impact paper that this aspires to be, the focus needs to be on the results of this exploratory approach. Instead, even after a lengthy introductory section, we still don't know what the authors would like us to take away as the major contribution of their work.

The Results section, in turn, is structured more as a narrative of the research process (“first we did X, then Y, then Z”) than a report of project findings. While this does obviously include the results themselves, it does not foreground the key findings in the way that I would expect, and since the data sources and modeling process have not been introduced, I am left wondering how/if they might have biased the results.

Fig. 3 seems to be the core result of the project, but it is not entirely clearly presented. If I understand correctly, the suitability maps presented here were calculated with reference to site locations in each period (i.e., site locations were used to identify the suitable values for the climatic and environmental variables used in each period). That being the case, these are effectively summaries of the kinds of areas inhabited by Neolithic farmers (because the constituent site location data haven't

really been presented in any detail, it's hard to assess whether taphonomy or other biases might be partly responsible for structuring this).

With a model structured in this way, it's not possible to explore whether changes in settlement distributions are due to climate-driven changes in the locations suitable for Neolithic agriculture, or instead to other factors. This would I think require an independent *crop suitability* model based on plant physiology rather than locations of agricultural settlements. In the absence of the ability to address that question, though, I'm left wondering just what we can *do* with the model results.

Conclusions

The conclusions are actually quite limited. The primary goal is succinctly defined:

"This study's major goal was to explore and evaluate the climatic and environmental factors that influenced the distribution of early Neolithic farmers and their adaptation strategies in a circumscribed study region and over a defined time span." (Lns 540-542) And indeed this has been accomplished, but the authors don't really offer a clear message about what we can *do* with the results.

A few specific comments:

p2

In my view the introduction should at least mention the issue of migration vs. adoption.

p2, Ln51

Not clear what is meant by, "with an unknown status".

p3, Ln74

"The literature has revealed" – passive construction. Who, and how?

The remainder of this paragraph (to Ln98) could be condensed.

p4, Ln99

The introduction of the study area would benefit from a map.

p5

Sources for the paleoclimate data? How accurate are they, and how generalizable across the study area? Those details are vital here, but not presented until later. Even a brief gloss and a reference to the subsequent section would be helpful here.

p6

Critical here is where the archaeological survey data used comes from, how complete it is, what potential problems of taphonomic or research bias there (and whether they are uniform in time and space) – but none of that information is presented until much later.

Fig. 1

Wouldn't it make sense to include, for each time step, the values for site locations (i.e., a vertical distribution of points)? The principal argument seems to be that those should be tracking the climate changes.

p18

"The state of the art may influence the suitability models of some of the crops"

A bit unclear - I guess this refers to available data? But since the way in which the model has been constructed hasn't been presented yet, this caveat is hard to interpret.

p18

This forthright discussion of modeling limitations is very welcome - but because the modeling process hasn't been presented, we don't have any means to judge this.

p18

Difficult to assess this section too, when the data sources are not described until a subsequent section.

Section 4.4

Even if the details of the materials and methods are confined to this section, a gloss on them needs to come in the introduction in order for the rest of the study to make sense.

p26

"(1) early Neolithic farmers settled preferentially in ecological ecotones or niches where small-scale farming with domestic animals was also possible and (2) they abandoned overexploited areas for new territories with similar characteristics when these habitats and niches changed due to climatic factors (increases in temperature, changes in precipitation patterns and seasonality) towards higher altitudes in search for more suitable climatic conditions." (Ln558-563)

• Why "overexploited"? I understood the argument built on the modeling to be that Neolithic farmers pursued socioecological niches when those moved due to climate shifts. Overexploitation is a separate issue.

- Similarly, why “with domestic animals”? That aspect of Neolithic farming has not been mentioned previously and does not figure in the modeling process.

References mentioned:

Verhagen, Philip, Laure Nuninger, Frédérique Bertoncello, and Angelo Castorao Barba
2015 Estimating the “memory of landscape” to predict changes in archaeological settlement patterns. 43rd Computer Applications and Quantitative Methods in Archaeology Conference: 623–636.

Reviewer #2

(Remarks to the Author)

This exciting paper uses climatic variables, previously downscaled from an open-access dataset (CHELSA-TraCE21k d v.10), to re-interpret Neolithic settlement and archaeobotanical crop distributions in a western Mediterranean study region. The underlying hypothesis is that climatic variability across the study region and climate change through time shaped the distribution of Neolithic settlement and of Neolithic crops. This hypothesis is already generally accepted for Neolithisation in the western Mediterranean, in that for example early settlement focused on coastlines followed by shifts inland to cooler and wetter zones. Similar patterns of Neolithisation are known across the Mediterranean. What this new paper reveals, however, is that in the western Mediterranean the inland shift coincided with a long-term climatic trend towards warmer and less seasonally variable conditions. The implication is that farmers responded to climate change by moving to higher altitude, and not as a shift to specialisation in particular products, but rather to maintain the resilience of the whole mixed farming network.

A second dimension of this paper is the modeling of Neolithic crop distributions through time, using a database of radiocarbon-dated archaeobotanical charred crop remains. The aim here is to assess the ecological niches of individual crops. As for the climatic analysis of Neolithic settlement patterns, the necessary assumption is that these distributions are not biased taphonomically, but in the case of archaeobotanical crop distributions this is unlikely to be the case, as the authors point out in relation to the apparent upland distribution of opium poppy. For the different cereals a broad niche, with mild winters and low precipitation, is reconstructed. The fact that cereals are similarly distributed suggests that uniformitarian extrapolation of the present-day ecological preferences of different cereals, as set out in the introductory section of the paper, could be revisited. The authors conclude that some shifts in crop spectra can be interpreted as resilience to climate change, while others cannot be explained in this way and instead suggest cultural, culinary choices.

Overall, the argument of the paper, its underlying data and methodology are set out clearly, and the results are of wide relevance to studies of Neolithisation processes. I have two suggestions for minor revision. One is that the concept of ‘habitat suitability’ based on species distribution modeling be clearly defined in the main text, as an inference based on extrapolation from (here) archaeological site distributions, and not on a priori arguments about the suitability of different landscapes. A second suggestion is that the authors include commentary in the Discussion and/or Conclusions regarding the potential of climate modeling to reconstruct ancient crop ecologies and hence to refine the uniformitarian inferences we otherwise make from existing present-day crops. This is a potential for the future, as archaeobotanical and climate datasets gain higher resolution. Moreover, a climate model-based approach to ancient crop ecologies would be complemented by independent lines of evidence, including from stable isotope analysis of crop remains and ecological analysis of associated arable weed flora.

A minor bibliographic point is that the key citation to the downscaling of the CHELSA-TraCE21K dataset should be updated to Karger et al. 2023 (Karger, D. N., Nobis, M. P., Normand, S., Graham, C. H., and Zimmermann, N. E.: CHELSA-TraCE21k – high-resolution (1 km) downscaled transient temperature and precipitation data since the Last Glacial Maximum, *Clim. Past*, 19, 439–456, <https://doi.org/10.5194/cp-19-439-2023>, 2023).

Reviewer #3

(Remarks to the Author)

This paper aims at exploring in an explicit quantitative framework the interplay between early farming systems and climatic/environmental variables using a regional time transect. In this sense, it only addresses in a tangential way the notion of spread of agriculture (which would require, for instance, the same process to be done across several regions, then compared together). Although it draws on robust data and - somewhat- innovative methods, it presents several issues related to results, reporting the results, and their comparison with the existing state of the literature, so that it is not possible to recommend it for publication in the present format.

The introduction is relatively poor: the opening paragraph sounds like a statement regarding the relationship of early farming and climate, which is surprising as it seems to be actually the research question of the paper, as explicitly framed as a hypothesis on paragraph starting line 112 (plus contradiction with 81-3). The first paragraph overall is very generic, by contrast with the far too precise second paragraph. Otherwise, the text starting line 86 on the relevance of archaeological knowledge for future sustainability issues is sound, but effectively very cliché (as are the eventual conclusions actually). Another issue with the introduction lies in the way it sets up the methodological landscape: most of the references deal with the Pleistocene, which indeed has been a focus period for the use of such tools, but overlook some key dimensions of the literature. This problem recurs across the paper (see below), and here, I would refer solely to the, non-quoted, work by Vidal-Cordasco and Nuevo-Lopez on niche construction in the Early Neolithic in Iberia (Vidal-Cordasco, M., & Nuevo-

López, A. (2021). Difference in ecological niche breadth between Mesolithic and Early Neolithic groups in Iberia. *Journal of Archaeological Science: Reports*, 35, 102728. <https://doi.org/10.1016/j.jasrep.2020.102728>. In a related way, in the discussion (lines 380 and following), suggested references are of limited relevance and/or very general, whilst specific literature on SDm and the European Neolithic is surprisingly absent (e.g. Banks, W. E., Antunes, N., Rigaud, S., & Francesco d'Errico. (2013). Ecological constraints on the first prehistoric farmers in Europe. *Journal of Archaeological Science*, 40(6), 2746–2753. <https://doi.org/10.1016/j.jas.2013.02.013>

Sánchez Goñi, M. F., Ortu, E., Banks, W. E., Giraudeau, J., Leroyer, C., & Hanquiez, V. (2016). The expansion of Central and Northern European Neolithic populations was associated with a multi-century warm winter and wetter climate. *Holocene*, 26(8), 1188–1199. <https://doi.org/10.1177/0959683616638435>).

On a purely editorial note, the references are listed in alphabetical order rather than in order of call in the paper; not only is this at odds with referencing systems in *Nature Communications* (and most journals), but this proves really unhelpful for reviewing and assessing the quality and relevance of the bibliography.

From a methodological point of view, the paper is also not exempt of problems. Firstly, the description of the dataset is very limited: in total, how many sites were effectively used? I guess 800+, but the total absence of any form of sensitivity analysis is damaging and, frankly, not acceptable: are there any spatial trends/biases in the distribution of the sites? what are exactly the taxa recorded (using presence/absence I guess)? What factors shape the archaeobotanical dataset, especially regarding the way archaeobotanists have worked? The lack of any assessment of this is damageable as 1) there is some existing literature on such dimensions, 2) it would allow to address some of the points made - once more in very generic - terms in the discussion (see lines 322 and following). For instance, on lines 257 and following, I would like to know the implications of varying sample sizes between periods (see also fig. 5), but am also wondering if there is any influence / pattern in degrees of associations between different taxa (were these independently tested for instance?)

Further, the dataset is split into four chronological bins, but with very limited justification as to why: given that these have different durations, I expect that they also reflect some sort of cultural component. This needs to be explicitly mentioned AND evaluated given how instrumental this division is for the entire analysis. Would using artificial chronological brackets lead to different results? Also, are they any general pattern when comparing these four brackets, for instance in terms of number of sites? Are they any biases at play here?

The authors use modelled climatic data, which is not a surprise, but never really justify the rationale behind the 18 reconstructed variables: aside from considering that climate matters, I would have expected some discussion here. There might not be a need to discuss all 18 of them, as the authors rightly check for autocorrelations, but suggestions / hypotheses as to the implications of these variables for plant distributions would be welcome, rather than a strict quantitative discussion. Also, I note that, in the methods section, Bio01 has been discarded, though it features extensively in the results and discussion; this looks like a contradiction and ought to be either corrected or fully explained.

As for topographic variables, the authors use elevation and slope, but do not check for autocorrelation between them, which seem a bit hasty. Also, and in view of the SDM literature in ecology, why did the authors not explore / use other topographic variables, such as terrain ruggedness index? Regarding distance to water, more precision is needed to the underlying dataset used? Also, what does "important water resources" (line 439) effectively encompass?

Regarding the results, the comparison between different techniques and periods is potentially exciting (e.g. lines 204-5), but it is unclear to me how this comparison was effectively performed: is this simply a question of visual inspection, or were any formal / quantitative tests applied? In a related way, the text highlights that both techniques provide relatively comparable distributions, though very different predictor values; this is either very exciting (as it would point to how complimentary the techniques are) or very concerning (as it could suggest that the methods simply identify a distribution actually driven by a lurking, non-explored, variable; or simply that the results are mere methodological artefacts of different techniques), but in any case this requires more attention.

Data treatment and visualisation is also sometimes not coherent: for instance, on fig. 6, it is unclear what is gained by using a 2d kde to visualise the data? More fundamentally, why using here a continuous chronological x axis (and how is this done? median of 14C dates; what if there are several dates per assemblages?), rather than a discrete classification in four phases as per the rest of the paper? Since this is a 2d kde, effectively variation in the x axis merely maps temporal variation, and I am left wondering if we are not simply here looking at the structure of the dataset, rather than any meaningful pattern (e.g. how does the distribution of dated archaeobotanical assemblages compare to the total number of dated sites?).

Regardless of the aforementioned problems with the lack of sensitivity analysis and other methodological points raised, my main issue with the discussion lies in its very limited engagement with the existing literature. Aside from the previously quoted paper on niche modelling for the Iberian Neolithic, there is quite a lot of literature on the change in settlement pattern during the Neolithic in the area (e.g. Perrin, T., Manen, C., Valdeyron, N., & Guilaine, J. (2018). Beyond the sea... The Neolithic transition in the southwest of France. *Quaternary International: The Journal of the International Union for Quaternary Research*, 470, 318–332. <https://doi.org/10.1016/j.quaint.2017.05.027>; on a more archaeobotanical point of view: see also Bouby, L., Marival, P., Durand, F., Figueiral, I., Briois, F., Martzluff, M., Perrin, T., Valdeyron, N., Vaquer, J., Guilaine, J., & Manen, C. (2020). Early Neolithic (ca. 5850-4500 cal BC) agricultural diffusion in the Western Mediterranean: An update of archaeobotanical data in SW France. *PLoS One*, 15(4), e0230731. <https://doi.org/10.1371/journal.pone.0230731>).

Admittedly, this literature is based on qualitative assessment of the data, but I would expect these to be discussed, and their results to be evaluated in view of the methodology adopted here. In this sense, I have extensive reservation as to the originality of the results presented here.

Version 1:

Reviewer comments:

Reviewer #1

(Remarks to the Author)

The authors have revised the manuscript significantly, and to good effect: the study and its results are presented with much greater clarity. However, it is still quite long and discursive, more focused on describing the process of this study than making an argument based on the results.

I would suggest that for a big-data archaeological modeling paper to be effective it should do three things: 1) make the case that the data are of sufficient quality that the results should be reliable, 2) argue that the methods employed are appropriate to the questions under consideration, and 3) make a significant claim about the human past.

Following the revisions, this paper does a thorough job with (2), but doesn't really consider (1) in detail (though it is acknowledged) and never really gets around to (3) (ironically, since I think there is considerable potential to do make interesting and significant claims).

There is a clear statement of purpose at the beginning of the Conclusion (p31): "The main objective of this research was to provide a first characterization for both the ecological and the environmental niche of Neolithic settlements and their food plant resources, cultivation practices and variations in time and space across the Western Mediterranean area, based on a big database and a data-driven approach using machine learning techniques." This *should* be (3) - but the paper never provides a simple statement of what those occupied niches are, and how they compare to the niches of particular food plants! I think that it could, and that that would provide the summary message that is not so far clearly articulated. If I follow the modeling process correctly (p25), it seems that two distinct datasets have been analyzed: one of site locations, and one of site locations where particular crops have been found. If that's the case, it seems to me that an obvious question would be how the observed niches based on site distribution for a given time period relate to the observed trans-temporal crop niches. That is, if - based on all occurrences (it sounds like the data are presence/absence rather than abundance?) of, e.g., chickpeas - it is possible to define a chickpea niche, then that niche might be compared to the one occupied by Neolithic farmers for any given time period.

This would provide a means of *using* the model results. The four conclusions on p32, in contrast - while interesting and thoughtful - seem to have more to do with general observations of the datasets compiled than they do with the modeling results. That being the case, the claim that follows - "the combined analyses performed in this paper prove extremely useful to answer questions regarding agricultural decision-making and spreading in the past" - doesn't seem to follow from the paper. I don't think that this is necessarily a shortcoming of the study - it may be the inevitable outcome of uncertainty in the data and ambiguity in the results - but it is a problem that could be avoided with a clearer focus on a few key things that the authors have learned about the Neolithic.

Finally, landscape taphonomy (patterns of site preservation and discovery) and site taphonomy (patterns of preservation of archaeobotanical remains) are both of great importance here, as are data biases more generally. There is some explicit recognition that these may exist (e.g., p21), but not enough consideration of the ways in which these might affect the dataset, analyses, and subsequent conclusions.

Specific comments on the text:

Ln66-68

"The main hypothesis of the paper is that choices regarding site location and crop cultivation made in periods of climatic change most likely reflect adaptive strategies of prehistoric populations."

Is this the hypothesis of the paper, or an assumption underlying the analysis? This is not really a hypothesis that the paper tests, but rather an underlying premise: by looking at site locations we can garner information about adaptive strategies.

Ln77-82

Does this abundance of data on pile dwelling spatially bias the dataset towards some areas at the expense of others?

More generally, it would be important to recognize in the Introduction that an HS approach based on archaeological settlement data is vulnerable to taphonomic and recovery biases. Perhaps those don't matter here, but that's an argument that needs to be made.

Ln164-166

"These changes are unlikely to be the result of a climate change that would make these regions suddenly fall into the suitability area of these crops."

Ok - why is that unlikely? Isn't determining that likelihood one of the objects of the modeling described here?

p6, Lns 176-181

"The aims of the paper are to 1) explore and reveal possible climatic dynamics in the study area; 2) examine the settings of human and crop niches; 3) evaluate landscape/climate-level factors that may have contributed to the crops shifting or to maintaining these distributions over space and time; 4) while mapping potential suitable areas of early farmers' settlements; 5) ultimately, to assess the capacity of these techniques to address relevant archaeological and archaeobotanical questions."

The way these aims are framed raises some questions. With respect to (1), isn't climate data an input, not an output? With respect to (2) and (3) - isn't there a more basic question here, which is simply that of determining whether the niches occupied by Neolithic farmers changed over time? From that follow others: did niches move in space following climate changes, but remain stable in character? If they changed in character, was that related to the adoption of different crops? There are a lot of interesting questions with which this research is engaged but which aren't quite articulated clearly enough.

here.

Item (4) is not an aim – it needs a verb, and (5) would be more effective if it were more specific about what the relevant questions were. Aim (5) also raises the question of whether the paper's ultimate goal to assess the techniques, or to say something about Neolithic farming?

Fig. 1

This map raises the question of data biases, addressing which is, I would argue, fundamental to the project. Surely part of the story of this map is that sites are scarce in the Po Valley at least in part for reasons of landscape taphonomy. This has been the subject of some attention, e.g. recently:

Marco Cavalazzi. (2021). "Looking through the keyhole": problems and research strategies for landscape archaeology in an alluvial plain with a high rate of vertical growth. The case of Bassa Romagna and south-eastern Po Valley. *GROMA: Documenting Archaeology*, 5. <https://doi.org/10.32028/Groma-Issue-5-2020-1358>

That provides the opportunity to bring up the issue, and argue that it does not compromise the results. Perhaps the patterns that the authors detect are so robust that biases are irrelevant, but that's an argument that has to be made.

Fig. 2

These are, I guess, CHELSA Trace21k data spatially averaged over the study area? That should be made clear in the caption.

Ln198-201

"four Phases are based on crop dynamics and divide our study period according to changing dominance of crop assemblages"

This is an interesting approach (and possibly a paper in its own right!). How does the division compare with traditional culture historical divisions?

Fig 3

Caption mentions boxplots, but there are no boxplots in the figure.

Ln265

"absence of suitable areas in the Po valley"

This is why the potential biases in the data are an issue. If there are sites in the Po Valley but they're underrepresented, then those areas will necessarily be classified as low suitability.

Ln271-273

Sentence fragment. And an interesting claim - why not?

Ln334-336

If I understand correctly, this is based on presence of particular crops at various sites/phases. That being the case - different issues of bias/taphonomy to site preservation/dating need to be considered (as indeed is noted below on p21).

Fig.7

A qualitative color palette would be more appropriate than a ramp here, since these are discrete values rather than a continuum.

p22

"...the distribution of Neolithic settlements and the main cultivated crops needed to overcome different adaptive challenges in different zones of the Western Mediterranean area and that these challenges varied over time."

This might be true – and is an interesting claim – but doesn't seem to me like a direct outcome of the model. I think it would be more appropriate as an interpretation to be argued.

Ln449-453

"With some exception, the differences in climatic niches for barley, glume wheats and naked wheats are less relevant, which could point at farmer's choices and cultural influences as determining factors rather than ecological ones, at least when considering broad patterns (local or regional studies might be more suited to identify climate-related agricultural decision-making)."

Unclear. Why less relevant?

Ln631

"numbtechnique"?

Ln669 & 674

"predominant taxa" and "gain importance"

Maybe this is an outcome of the compilation of the archaeobotanical database that's described above, but is not so far as I can tell an outcome of the modeling. In fact I don't think there's been any discussion of the relative abundance of different taxa in time/space.

Ln679-680

"farmers responded to climate change by moving to higher altitude (broadening their niche)"

Is this niche expansion associated with population growth? Of course population proxies present a difficult problem in their own right, and one perhaps beyond the scope of this study, but there's an important question here that gets at the issue of adaptation on which the authors focus: are farmers moving in order to continue occupying the same niche space as climate change causes shifts in niche locations (i.e. mobility as adaptation), or are farmers moving for other reasons (e.g. population growth) and necessarily having to adapt to the new niches in which they find themselves? These are very different scenarios, and to my mind demonstrate that simply appealing to "adaptation" is too general.

(Remarks on code availability)

Reviewer #3

(Remarks to the Author)

I thank the authors for extensively rewriting the text and answering all my queries and those from the other reviewers

(Remarks on code availability)

Code not available - unless mistake of mine - in the package made available to the reviewers, and not publicly available yet on the github repository linked in the manuscript

Version 2:

Reviewer comments:

Reviewer #1

(Remarks to the Author)

The authors have responded to my comments in detail. I appreciate the close consideration that they have given them, and their willingness to engage with my comments, which have been very much intended to be constructive but which I'm sure were also a source of frustration.

(Remarks on code availability)

REVIEWER 1 COMMENTS

Blue : Response to reviewer comments

Reviewer #1 (Remarks to the Author):

The project described in this paper is interesting and ambitious, and appears to have been carried out with admirable thoughtfulness and attention to detail. I enjoyed learning something about the work, and see this kind of combination of data integration and modeling as an important and promising route forward for archaeology. However, I think that this paper needs substantial revision in order to be publishable. In its current form, I find the modeling process hard to follow, making the conclusions difficult to evaluate – and the conclusions themselves seem more focused on the successes in creating a model than on what can be learned from it.

On p25 the authors summarize the goal of the project: “the development of a quantitative data driven approach to model and approximate the features of the ecological niches exploited by the first Neolithic farmers.”

Conceptually this begins with locational modeling, i.e. characterizing the regions in which archaeological sites are found. Taking paleoclimate data into account provides a means of taking into account that the characteristics of these locations may have differed when they were selected as settlement locations. I see this as sound practice, and have done similar things myself. I might quibble over methodological details (the authors don’t seem to have taken into account issues of settlement persistence / path dependence, interestingly explored in e.g. Verhagen et al. 2016), but in fact I think that the exploration of RF and MaxEnt is a step forward in most respects. My concerns don’t have to do with the modeling per se but rather the way in which it is presented and the conclusions drawn from it.

We thank the reviewer for pointing at this issue and have rewritten and reorganized the text accordingly.

Presentation

The most succinct statement of the paper’s import does not come until p4 (Lns99-101), and then it is a statement of ambition rather than of findings:

“This paper aims at exploring the impact of environmental and climatic constraints on the distribution of the first farming populations in the Western Mediterranean area during the period of 5900-2300 cal. BC.”

This characteristic is repeated throughout the introduction: the paper is framed more as a grant proposal (“this is what we will attempt to do”) than a report of results (“this is what we found when we did X”). On Lns 117-118, for example (“In order to investigate such hypotheses, we developed an innovative multi-proxy and interdisciplinary approach”) and in the concluding lines (127-133) of the Introduction (where the verbs employed are “explore and reveal”, “examine”, “evaluate”, “map”, “assess”, “diagnose”). Of course doing these things is fundamental to scientific investigation, but for the purposes of the high-impact paper that this aspires to be, the focus needs to be on the results of this exploratory approach.

Instead, even after a lengthy introductory section, we still don't know what the authors would like us to take away as the major contribution of their work.

We thank the reviewer for pointing at this issue and have rewritten and reorganized the text accordingly.

The Results section, in turn, is structured more as a narrative of the research process ("first we did X, then Y, then Z") than a report of project findings. While this does obviously include the results themselves, it does not foreground the key findings in the way that I would expect, and since the data sources and modeling process have not been introduced, I am left wondering how/if they might have biased the results.

Fig. 3 seems to be the core result of the project, but it is not entirely clearly presented. If I understand correctly, the suitability maps presented here were calculated with reference to site locations in each period (i.e., site locations were used to identify the suitable values for the climatic and environmental variables used in each period). That being the case, these are effectively summaries of the kinds of areas inhabited by Neolithic farmers (because the constituent site location data haven't really been presented in any detail, it's hard to assess whether taphonomy or other biases might be partly responsible for structuring this).

We thank the reviewer for these comments. The text has now been improved and for better clarity Fig. 3 has been split in Fig. 4 and 5.

With a model structured in this way, it's not possible to explore whether changes in settlement distributions are due to climate-driven changes in the locations suitable for Neolithic agriculture, or instead to other factors. This would I think require an independent *crop suitability* model based on plant physiology rather than locations of agricultural settlements. In the absence of the ability to address that question, though, I'm left wondering just what we can *do* with the model results.

We are not trying to investigate climate-driven changes, since any attempt on this issue would now be pure speculation given the state of research in both palaeoclimatology and archaeology. We want to characterise the niche inhabited by farmers and see if changes in this niche mirror other changes. We are thankful to reviewer #2 for helping us in addressing reviewer #1 comment: "*The underlying hypothesis is that climatic variability across the study region and climate change through time shaped the distribution of Neolithic settlement and of Neolithic crops.*".

Conclusions

The conclusions are actually quite limited. The primary goal is succinctly defined: "This study's major goal was to explore and evaluate the climatic and environmental factors that influenced the distribution of early Neolithic farmers and their adaptation strategies in a circumscribed study region and over a defined time span." (Lns 540-542) And indeed this has

been accomplished, but the authors don't really offer a clear message about what we can *do* with the results.

The conclusions have been re-written, thanks to the reviewer's comments. Reviewer #2 kindly strengthens our major contributions: *"What this new paper reveals, however, is that in the western Mediterranean the inland shift coincided with a long-term climatic trend towards warmer and less seasonally variable conditions. The implication is that farmers responded to climate change by moving to higher altitude, and not as a shift to specialisation in particular products, but rather to maintain the resilience of the whole mixed farming network. [A] second dimension of [...] this paper is the modeling of Neolithic crop distributions through time, using a database of radiocarbon-dated archaeobotanical charred crop remains. The aim here is to assess the ecological niches of individual crops. [...]. For the different cereals a broad niche, with mild winters and low precipitation, is reconstructed.[...] The authors conclude that some shifts in crop spectra can be interpreted as resilience to climate change, while others cannot be explained in this way and instead suggest cultural, culinary choices.*

Furthermore, Reviewer#2 points at *"[...]the potential of climate modeling to reconstruct ancient crop ecologies and hence to refine the uniformitarian inferences we otherwise make from existing present-day crops. This is a potential for the future, as archaeobotanical and climate datasets gain higher resolution. Moreover, a climate model-based approach to ancient crop ecologies would be complemented by independent lines of evidence, including from stable isotope analysis of crop remains and ecological analysis of associated arable weed flora."*

A few specific comments:

p2

In my view the introduction should at least mention the issue of migration vs. adoption.

Thank you for the comment. We added relevant references for this issue, but in the way we are treating and interpreting our datasets, it is relatively irrelevant whether crops were adopted by local populations or whether they reflect migrations. We are interested in the niche where these crops are grown and how it changes over time.

p2, Ln51

Not clear what is meant by, "with an unknown status".

This has been corrected. Thank you for the comment.

p3, Ln74

"The literature has revealed" – passive construction. Who, and how?
The remainder of this paragraph (to Ln98) could be condensed.

This has been corrected. Thank you for the comment.

p4, Ln99

The introduction of the study area would benefit from a map.

We have added Fig. 1 showing the Study area with the ecological zones and the sites locations. Thank you for the comment.

p5

Sources for the paleoclimate data? How accurate are they, and how generalizable across the study area? Those details are vital here, but not presented until later. Even a brief gloss and a reference to the subsequent section would be helpful here.

Thank you for the comment. Due to the structure of this kind of paper, the data and the methodology are presented exhaustively only after the results and discussion, in sec. 4.2. In any case, this information has now been also included in the Introduction, for better clarity.

p6

Critical here is where the archaeological survey data used comes from, how complete it is, what potential problems of taphonomic or research bias there (and whether they are uniform in time and space) – but none of that information is presented until much later.

Thank you for the comment. Due to the structure of this kind of paper, this information about the data is exhaustively presented in section 4. This response also applies to comments on p18.

The datasets used within this research have been published and discussed previously by Antolin et al. 2018 [ref. 18]; Martinez-Grau et al. 2020 [ref. 15]; 2021 [ref. 10]; Jesus 2021 [ref. 70]; Jesus et al. 2021 [ref. 49]. The crops data are also discussed in Antolin & Jacomet 2015 [ref. 4]; Antolin 2016 [ref. 2]; Antolin et al. 2016 [ref. 3]; Antolin et al. 2021 [ref. 36]; Antolin et al. 2022 [ref. 20], Antolin 2024 [ref. 50]; Antolin & Schäfer 2024 [ref. 45] and these publications have now been presented in the Introduction and through the manuscript with cross-references for greater clarity.

Following the reviewer's comment, Fig. 7, that already showed the quantified crop occurrences per phase and density of crop occurrences over time, has been now modified to show the total number of crop occurrences per phase. We have also produced and added the new Fig. 10 in section 4.1, to provide a description of the number of sites/crops per phase, for better clarity.

As for the taphonomic and biases issues, we followed standard procedures of spatial data exploration (ESDA) as presented in Figs. 3, 7, 8 and 9 and the datasets used (especially the archaeological and the archaeobotanical) have already been analysed and presented in previous publications (see the refs. above), where also chronological differences were highlighted. There is nothing we can do at this moment to overcome these difficulties and, at

the same time, it is the first time that we have datasets of this quality and resolution for the study region, which justifies the explorative modelling approach presented here. In any case, we applied a filtering approach to reduce the negative effect of sampling bias in the modelling procedure, as described in detail in section 4.4. This approach applies the statistical technique called spatial k -fold cross validation, where the training and testing data were selected far enough apart in the geographic space, to reduce the number of presences in oversampled regions or oversampled environmental conditions in the environmental space.

Fig. 1

Wouldn't it make sense to include, for each time step, the values for site locations (i.e., a vertical distribution of points)? The principal argument seems to be that those should be tracking the climate changes.

We are thankful for this suggestion. We decided to keep the original Figure 1 as it was (now Figure 2), since it shows the results of the palaeoclimatic model developed by Karger et al. 2023 [ref. 12] applied to the spatio-temporal framework of our study area, but we changed Fig. 3 showing site location per Phase, as suggested by the reviewer. The supplementary information contains additional figures (Suppl. Fig. 3a to 3d in SI) showing site types distribution per Phase (time-step) according to each of the climatic and environmental variables considered.

p18

“The state of the art may influence the suitability models of some of the crops”

A bit unclear - I guess this refers to available data? But since the way in which the model has been constructed hasn't been presented yet, this caveat is hard to interpret.

Thank you for this suggestion. This has now been better formulated in the text.

p18

This forthright discussion of modeling limitations is very welcome - but because the modeling process hasn't been presented, we don't have any means to judge this.

Thank you for this comment. This has now been better formulated in the text.

p18

Difficult to assess this section too, when the data sources are not described until a subsequent section.

Thank you. This has now been better formulated in the text.

Section 4.4

Even if the details of the materials and methods are confined to this section, a gloss on them needs to come in the introduction in order for the rest of the study to make sense.

We have included methodological aspects in the introduction as suggested by the reviewer, yet avoiding repetitions as far as possible. Thank you for the comment.

p26

“(1) early Neolithic farmers settled preferentially in ecological ecotones or niches where small-scale farming with domestic animals was also possible and (2) they abandoned overexploited areas for new territories with similar characteristics when these habitats and niches changed due to climatic factors (increases in temperature, changes in precipitation patterns and seasonality) towards higher altitudes in search for more suitable climatic conditions.” (Ln558-563)

- Why “overexploited”? I understood the argument built on the modeling to be that Neolithic farmers pursued socioecological niches when those moved due to climate shifts. Overexploitation is a separate issue.
- Similarly, why “with domestic animals”? That aspect of Neolithic farming has not been mentioned previously and does not figure in the modeling process.

The introduction has been rewritten and reorganised following the reviewer’s comments.

References mentioned:

Verhagen, Philip, Laure Nuninger, Frédérique Bertoncello, and Angelo Castrorao Barba
2015 Estimating the “memory of landscape” to predict changes in archaeological settlement patterns. 43rd Computer Applications and Quantitative Methods in Archaeology Conference: 623–636.

REVIEWER 2 COMMENTS

Blue : Response to reviewer comments

Reviewer #2 (Remarks to the Author):

This exciting paper uses climatic variables, previously downscaled from an open-access dataset (CHELSA-TraCE21k d v.10), to re-interpret Neolithic settlement and archaeobotanical crop distributions in a western Mediterranean study region. The underlying hypothesis is that climatic variability across the study region and climate change through time shaped the distribution of Neolithic settlement and of Neolithic crops. This hypothesis is already generally accepted for Neolithisation in the western Mediterranean, in that for example early settlement focused on coastlines followed by shifts inland to cooler and wetter zones. Similar patterns of Neolithisation are known across the Mediterranean. What this new paper reveals, however, is that in the western Mediterranean the inland shift coincided with a long-term climatic trend towards warmer and less seasonally variable conditions. The implication is that farmers responded to climate change by moving to higher altitude, and not as a shift to specialisation in particular products, but rather to maintain the resilience of the whole mixed farming network.

A second dimension of this paper is the modeling of Neolithic crop distributions through time, using a database of radiocarbon-dated archaeobotanical charred crop remains. The aim here is to assess the ecological niches of individual crops. As for the climatic analysis of Neolithic settlement patterns, the necessary assumption is that these distributions are not biased taphonomically, but in the case of archaeobotanical crop distributions this is unlikely to be the case, as the authors point out in relation to the apparent upland distribution of opium poppy. For the different cereals a broad niche, with mild winters and low precipitation, is reconstructed. The fact that cereals are similarly distributed suggests that uniformitarian extrapolation of the present-day ecological preferences of different cereals, as set out in the introductory section of the paper, could be revisited. The authors conclude that some shifts in crop spectra can be interpreted as resilience to climate change, while others cannot be explained in this way and instead suggest cultural, culinary choices.

Overall, the argument of the paper, its underlying data and methodology are set out clearly, and the results are of wide relevance to studies of Neolithisation processes. I have two suggestions for minor revision. One is that the concept of ‘habitat suitability’ based on species distribution modeling be clearly defined in the main text, as an inference based on extrapolation from (here) archaeological site distributions, and not on a priori arguments about the suitability of different landscapes.

We appreciate this comment. The definition of “habitat suitability” is now explained in section 4.4.

A second suggestion is that the authors include commentary in the Discussion and/or Conclusions regarding the potential of climate modeling to reconstruct ancient crop ecologies and hence to refine the uniformitarian inferences we otherwise make from existing present-day crops. This is a potential for the future, as archaeobotanical and climate datasets gain

higher resolution. Moreover, a climate model-based approach to ancient crop ecologies would be complemented by independent lines of evidence, including from stable isotope analysis of crop remains and ecological analysis of associated arable weed flora.

We appreciate this comment. This has now been included in the conclusions.

A minor bibliographic point is that the key citation to the downscaling of the CHELSA-TraCE21K dataset should be updated to Karger et al. 2023 (Karger, D. N., Nobis, M. P., Normand, S., Graham, C. H., and Zimmermann, N. E.: CHELSA-TraCE21k – high-resolution (1 km) downscaled transient temperature and precipitation data since the Last Glacial Maximum, *Clim. Past*, 19, 439–456, <https://doi.org/10.5194/cp-19-439-2023>, 2023).

We thank the reviewer for this suggestion.

REVIEWER 3 COMMENTS

Blue : Response to reviewer comments

Reviewer #3 (Remarks to the Author):

This paper aims at exploring in an explicit quantitative framework the interplay between early farming systems and climatic/environmental variables using a regional time transect. In this sense, it only addresses in a tangential way the notion of spread of agriculture (which would require, for instance, the same process to be done across several regions, then compared together). Although it draws on robust data and - somewhat- innovative methods, it presents several issues related to results, reporting the results, and their comparison with the existing state of the literature, so that it is not possible to recommend it for publication in the present format.

The introduction is relatively poor: the opening paragraph sounds like a statement regarding the relationship of early farming and climate, which is surprising as it seems to be actually the research question of the paper, as explicitly framed as a hypothesis on paragraph starting line 112 (plus contradiction with 81-3). The first paragraph overall is very generic, by contrast with the far too precise second paragraph. Otherwise, the text starting line 86 on the relevance of archaeological knowledge for future sustainability issues is sound, but effectively very cliché (as are the eventual conclusions actually).

The whole structure, with the introduction, the results, the discussion and conclusion, has been rewritten and reorganised following the reviewer's comments.

Another issue with the introduction lies in the way it sets up the methodological landscape: most of the references deal with the Pleistocene, which indeed has been a focus period for the use of such tools, but overlook some key dimensions of the literature. This problem recurs across the paper (see below), and here, I would refer solely to the, non-quoted, work by Vidal-Cordasco and Nuevo-Lopez on niche construction in the Early Neolithic in Iberia (Vidal-Cordasco, M., & Nuevo-López, A. (2021). Difference in ecological niche breadth between Mesolithic and Early Neolithic groups in Iberia. *Journal of Archaeological Science: Reports*, 35, 102728. <https://doi.org/10.1016/j.jasrep.2020.102728>). In a related way, in the discussion (lines 380 and following), suggested references are of limited relevance and/or very general, whilst specific literature on SDM and the European Neolithic is surprisingly absent (e.g. Banks, W. E., Antunes, N., Rigaud, S., & Francesco d'Errico. (2013). Ecological constraints on the first prehistoric farmers in Europe. *Journal of Archaeological Science*, 40(6), 2746–2753. <https://doi.org/10.1016/j.jas.2013.02.013> Sánchez Goñi, M. F., Ortu, E., Banks, W. E., Giraudeau, J., Leroyer, C., & Hanquiez, V. (2016). The expansion of Central and Northern European Neolithic populations was associated with a multi-century warm winter and wetter climate. *Holocene*, 26(8), 1188–1199. <https://doi.org/10.1177/0959683616638435>).

Thank you for this comment. The Introduction, as well as the entire text, has now been reformulated following the Reviewers' comments. In regard to SDM literature, the most up-to-date literature on SDM in ecology, biogeography and conservation science (disciplines in which SDM find their origins) has been taken into account, in our view. We have also

considered the scarce existing literature on applied quantitative methods in archaeobotany for the period of interest. We therefore appreciate the references suggested and have included and discussed them where appropriate.

On a purely editorial note, the references are listed in alphabetical order rather than in order of call in the paper; not only is this at odds with referencing systems in Nature Communications (and most journals), but this proves really unhelpful for reviewing and assessing the quality and relevance of the bibliography.

Thank you for the comment. This issue has been corrected.

From a methodological point of view, the paper is also not exempt of problems. Firstly, the description of the dataset is very limited: in total, how many sites were effectively used? I guess 800+, but the total absence of any form of sensitivity analysis is damaging and, frankly, not acceptable: are there any spatial trends/biases in the distribution of the sites?

A glossy on the datasets used is now provided also in the Introduction, with cross-references for better clarity, as well as a more in depth description provided in the section 4. Materials and Methods and 4.1 Archaeological and Archaeobotanical data. The spatial exploration of the site and crop datasets (ESDA) has been performed and described in section 2. Results, and is further supported by Figures 3a to 3d and Figures 8a to 8w in the Supplementary Information (SI).

what are exactly the taxa recorded (using presence/absence I guess)?

The archaeobotanical dataset is a presence/absence dataset. We have added further details in section 4. Materials and Methods, and subsections: 4.1- 4.2, as well as in SI, with the Suppl. Note 1.

What factors shape the archaeobotanical dataset, especially regarding the way archaeobotanists have worked?

Following the Reviewers comment, we have added more details and cross-references also in the Introduction for greater clarity. However, the scope of this paper is not to present new, unpublished source datasets. The archaeobotanical dataset was generated and previously described in detail (including factors shaping the dataset), by Antolin et al. 2018 [ref. 18]; Martinez-Grau et al. 2020 [ref. 15]; 2021 [ref. 10]; Jesus 2021 [ref. 70]; Jesus et al. 2021 [ref. 49]. The crops data are also discussed in Antolin & Jacomet 2015 [ref. 4]; Antolin 2016 [ref. 2]; Antolin et al. 2016 [ref. 3]; Antolin et al. 2021 [ref. 36]; Antolin et al. 2022 [ref. 20], Antolin 2024 [ref. 50]; Antolin & Schäfer 2024 [ref. 45]. These works were referenced throughout the original version of the manuscript (except for the most recent publication: ref. 45, that has now also been included). For better clarity, section 4 is fully dedicated to the description of the material used in this study.

The lack of any assessment of this is damageable as 1) there is some existing literature on such dimensions, 2) it would allow to address some of the points made - once more in very generic - terms in the discussion (see lines 322 and following). For instance, on lines 257 and following, I would like to know the implications of varying sample sizes between periods (see also fig. 5), but am also wondering if there is any influence / pattern in degrees of associations between different taxa (were these independently tested for instance?) Further, the dataset is split into four chronological bins, but with very limited justification as to why: given that these have different durations, I expect that they also reflect some sort of cultural component. This needs to be explicitly mentioned AND evaluated given how instrumental this division is for the entire analysis. Would using artificial chronological brackets lead to different results? Also, are there any general pattern when comparing these four brackets, for instance in terms of number of sites? Are there any biases at play here?

Following the Reviewers' comments, the discussion, as the entire text, have been reformulated and improved. In our view, the most up-to-date literature in SDM for ecology, biogeography and conservation science has been taken into account.

The splitting of the data into 4 bins is based on the previous work performed by the AgriChange research group and the site and crop datasets have been described in Martinez-Grau et al. 2021 [ref.10], Jesus 2021 [ref. 70] and Antolin et al. 2021 [ref. 36], on which this study builds up particularly. To best address this comment, we have added further information in the Suppl. Note 1 of the SI.

As the 4 Phases/bins contain a different number of sites and crop presences (details can be found also in section 4) this can of course influence the model results.

In general, any changes to the dataset may generate different results. We, as archaeologists, analyse the data in an archaeologically meaningful way. The datasets are out there for anybody to perform any other analyses they may wish. But how can one interpret any results without an archaeological context that gives meaning to them? Would one look at Roman data without considering the political changes? It would be a completely different research question in any case and beyond the aim of this research paper.

The authors use modelled climatic data, which is not a surprise, but never really justify the rationale behind the 18 reconstructed variables: aside from considering that climate matters, I would have expected some discussion here. There might not be a need to discuss all 18 of them, as the authors rightly check for autocorrelations, but suggestions / hypotheses as to the implications of these variables for plant distributions would be welcome, rather than a strict quantitative discussion.

Thank you for the comment. We believe, that it is indeed a surprise that we use modelled paleo-climatic data from the Chelsa-Trace 21K dataset published by Karger et al. 2023 [ref. 12], because this dataset did not exist until very recently for the chronology we are working on and at the time step of 100 years. Following the reviewer's comment, we have now added a qualitative discussion about the implications of these variables for plant distributions in section 3.

Also, I note that, in the methods section, Bio01 has been discarded, though it features extensively in the results and discussion; this looks like a contradiction and ought to be either corrected or fully explained.

Thanks for the comment. Bio01 has been discarded in the modelling procedure because the modelling approach we chose to follow is data driven. According to the Spearman correlation test we performed (Fig. 9 in Supplementary information), this variable was highly correlated with Bio 01, 05, 06, 10 and 11, and therefore Bio01 was discarded, as Bio05 was less correlated to all other variables than Bio01. However, to the best of our knowledge, Bio01 (Mean Annual Temperature) is a variable easily understood intuitively, that's why we decided to discuss it even though it is not used for training the model. Furthermore, as stated at line 565 of the manuscript, our modelling pipeline follows the research workflow defined by the most recent literature (see the references in the manuscript) as explained in section 4, sub. sec. 4.4.

As for topographic variables, the authors use elevation and slope, but do not check for autocorrelation between them, which seem a bit hasty.

Nothing in this paper was done in a hasty way but has been carried out “*..with admirable thoughtfulness and attention to detail.*” (Rev. #1). It is the outcome of years of data mining and data processing and months of writing.

As suggested by Reviewer #3, we have tested the topographic variables correlation. Slope and elevation are correlated by less than 0.005 more than the threshold of 0.75, as shown in the Suppl. Fig. 9 added in the SI. Nonetheless, we decided to keep these variables as we considered them important factors to detect relations between site locations and their environmental surroundings, at play both in past human mobility choices and crop adaptations strategies.

Also, and in view of the SDM literature in ecology, why did the authors not explore / use other topographic variables, such as terrain ruggedness index?

In the view of the SDM literature in ecology and biogeography, we have included topographic variables, such as DEM and Slope (see comment above). As suggested by Reviewer #3, we have now also calculated the Terrain Ruggedness Index - TRI (see Suppl. Tab. 4 in SI) and the spatial correlation between the environmental variables (see Suppl. Fig. 9 in SI). As a result, TRI was not retained as modelling input.

Regarding distance to water, more precision is needed to the underlying dataset used? Also, what does "important water resources" (line 439) effectively encompass?

Thank you for the comment. Details on this have been added to section 4. Only the perennial rivers and lakes have been used for the modelling. Further details on these variables can be found in the added reference Martinez-Grau et al. 2021 [ref. 10].

Regarding the results, the comparison between different techniques and periods is potentially exciting (e.g. lines 204-5), but it is unclear to me how this comparison was effectively performed: is this simply a question of visual inspection, or were any formal / quantitative

tests applied? In a related way, the text highlights that both techniques provide relatively comparable distributions, though very different predictor values; this is either very exciting (as it would point to how complimentary the techniques are) or very concerning (as it could suggest that the methods simply identify a distribution actually driven by a lurking, non-explored, variable; or simply that the results are mere methodological artefacts of different techniques), but in any case this requires more attention.

Thank you for the comment. The comparison between the two different approaches tested was accurately performed based on the existing literature on Machine Learning methods and as such, in our study it follows several steps described through the manuscript, and in particular in the section 4. Materials and Methods. In the first version of the manuscript the comparison was presented within Table 4 of the Supplementary information document, titled: “AUC for each model and phase”. In the current version of the manuscript, we have deleted this Table and added a new Fig. 11 in the manuscript, to better present the comparison.

To summarise, the models evaluation and assessment in our manuscript follow these steps:

1. According to the literature (see for example the references cited in sec. 4.4), and during the data preprocessing, the dataset of the sites was split into training and testing set folds. We used a larger portion for training (4 folds) the models and the rest (1 fold) for testing the models' performance.
2. Both models were trained using only the training dataset along with background points (MaxEnt) and pseudo-absences (RF). We adjusted hyperparameters to optimise the procedure (e.g., the number of model iterations was set to 1000).
3. In the model assessment and evaluation part (the newly added section 4.5), we computed an appropriate evaluation metric to gauge and compare model performance (accuracy, sensitivity and specificity): the area under the ROC curve (AUC). This measure is independent of threshold selection making it a powerful tool for assessing model performance. We performed *k*-fold cross-validation to ensure the models' robustness and mitigate overfitting (5 folds and block size of 170'000 m). We examined and compared the model's predictions on the test dataset, which was not used for training.
4. In section 4.5, we also assessed and compared the importance of each variable for each computed model and for each of the Phases, to understand which variables had the biggest impact on the prediction and distribution of suitable areas. For RF, we calculated this with the statistical measure of Mean Decrease in Accuracy (Suppl. Fig. 4 in SI) which is a method of computing the feature importance based on permuted out-of-bag (OOB) samples (Han et al 2016). In order to provide a more easily understandable output for the model's behaviour and the relative importance of different variables in predicting outcomes, we have also added in the SI. Suppl. Fig. 5a to 5d that shows the Partial dependence plots (PDPs). PDPs assess the marginal effect of a variable on the predicted outcome while accounting for the average effect of all other variables in the model. For MaxEnt, we have computed and included the response curve plots (Suppl. Fig. 6a to 6d in SI: Response Plots for MaxEnt) where each response plot indicates the role of a particular variable depicted as a response curve showing the predicted relative occurrence rate (the suitable areas) against the values of that predictor (variable).

5. Finally, we discussed the model results in section 3, where we have further incorporated our domain knowledge to interpret the models outcomes, to assess if the predictions produced align with our knowledge and/or other factors affecting the site locations. We further performed a comparison of prediction statistics in Fig. 4 (now Fig. 6) and Suppl. Fig. 8 to 11 in SI of the previous version of the manuscript. For better clarity, Suppl. Fig. 8 to 11 have been replaced and the results summarised in the new Suppl. Fig. 7 included in the SI. One can consider RF and MaxEnt as complementary methods as RF is an ensemble learning method based on decision trees, constructing multiple decision trees during training that outputs the average prediction of the individual trees. MaxEnt however is a probabilistic modelling approach that aims to find the distribution of maximum entropy given a set of constraints. RF tends to capture complex interactions between variables, while MaxEnt might provide more generalised response curves (see sec. 4.4 for references and more details on the algorithms functioning). We believe that comparing the two techniques allows us to take advantage of the strength of the two models and to be more confident in the comparable predictions (as done for example by Zhao et al. 2022 [ref. 90], etc.).

Data treatment and visualisation is also sometimes not coherent: for instance, on fig. 6, it is unclear what is gained by using a 2d kde to visualise the data? More fundamentally, why using here a continuous chronological x axis (and how is this done? median of 14C dates; what if there are several dates per assemblages?), rather than a discrete classification in four phases as per the rest of the paper? Since this is a 2d kde, effectively variation in the x axis merely maps temporal variation, and I am left wondering if we are not simply here looking at the structure of the dataset, rather than any meaningful pattern (e.g. how does the distribution of dated archaeobotanical assemblages compare to the total number of dated sites?).

We thank the reviewer for this suggestion and decided to change Fig. 6 accordingly. Now, the related Fig. 8 and 9 in the manuscript show the 2d KDE over two variables (Bio01 and Bio12, Bio11 and Bio19). This visualisation allows us to show the relationship between two variables in a more effective manner, compared to different diagrams. To further address the reviewer's comment, we also added a complete list of figures (Suppl. Fig 8a to 8w in SI) showing the distribution of crop occurrences over each variable and for each Phase, as well as Suppl. Tab. 3. (in SI) showing the total number and relative frequency in percentage of each crop type. For a detailed description of the archaeobotanical source please see the refs.: Antolin et al. 2021, Jesus et al. 2021; Antolin et al. 2024.

Regardless of the aforementioned problems with the lack of sensitivity analysis and other methodological points raised, my main issue with the discussion lies in its very limited engagement with the existing literature. Aside from the previously quoted paper on niche modelling for the Iberian Neolithic, there is quite a lot of literature on the change in settlement pattern during the Neolithic in the area (e.g. Perrin, T., Manen, C., Valdeyron, N., & Guilaine, J. (2018). Beyond the sea... The Neolithic transition in the southwest of France. *Quaternary International: The Journal of the International Union for Quaternary Research*, 470, 318–332. <https://doi.org/10.1016/j.quaint.2017.05.027>; on a more archaeobotanical point of view: see also Bouby, L., Marinval, P., Durand, F., Figueiral, I., Briois, F., Martzluff, M.,

Perrin, T., Valdeyron, N., Vaquer, J., Guilaine, J., & Manen, C. (2020). Early Neolithic (ca. 5850-4500 cal BC) agricultural diffusion in the Western Mediterranean: An update of archaeobotanical data in SW France. *PloS One*, 15(4), e0230731.

<https://doi.org/10.1371/journal.pone.0230731>). Admittedly, this literature is based on qualitative assessment of the data, but I would expect these to be discussed, and their results to be evaluated in view of the methodology adopted here. In this sense, I have extensive reservation as to the originality of the results presented here.

Thank you for this comment. The discussion, as well as the entire text, has now been reformulated following the Reviewers' comments. We appreciate the references suggested and have included and discussed them where appropriate. We would like to stress on the main aim of our research, which is not simply studying the change in settlement pattern during the Neolithic in the area, but the change in agricultural practises between 5900-2300 cal. BC over the wide Western Mediterranean area (gathering an increased and heterogenous amount of data, from archaeological/archaeobotanical to paleoclimate models) led by climatic and environmental conditions, and by using machine learning techniques to characterize and better understand the human and crop niches. As suggested by Rev.#2: “[A] second dimension of [...] this paper is the modeling of Neolithic crop distributions through time, using a database of radiocarbon-dated archaeobotanical charred crop remains. The aim here is to assess the ecological niches of individual crops. [...] For the different cereals a broad niche, with mild winters and low precipitation, is reconstructed.” Furthermore, Rev.#2 points at “[...]the potential of climate modeling to reconstruct ancient crop ecologies and hence to refine the uniformitarian inferences we otherwise make from existing present-day crops.” Quantitative attempts to reconstruct changes in the agricultural practices, combining archaeological, archaeobotanical and paleo-climate data, for the Neolithic (5900-2300 cal. BC), in the extended Western Mediterranean area, are only a few. Indeed, until recently, agricultural practices in our study area were known at a very coarse level and such problem was mainly due to the lack of archaeobotanical sample availability per phase and of isotope data (for Catalonia in the fourth millennium BC, and for France in the fifth millennium BC; see for example the recently published work of Antolin. et al. 2023 on this point). The two references suggested above (Banks et al. 2013 and Sanchez-Goni et al. 2016) date back to about 10 years ago, when the potential of machine learning techniques was not yet widely exploited in archaeological / archaeobotanical research. At the time, archaeological / archaeobotanical research did not have yet large datasets of surely dated archaeobotanical samples (as we have today) available to analyse supra-regional case-studies, on extended temporal scale. Moreover, these early attempts could also not combine yet the scanty archaeobotanical datasets with the high-resolution paleo-climatic models' outputs (such as the one produced by Karger et al. 2023, that provides today information for every 100 years for the last 21K years) having been the paleo-climate model published for the first time (as pre-print) only in 2021. It is clear that more data was needed.

Another major difference, between those referenced works and ours, lies mainly in the fact that none of them and of the examples constructed in archaeology so far, at the best of our knowledge, performs and compares two different techniques such as MaxEnt with RF. The case studies available in the current archaeological literature, as well as those further suggested by Rev. #3, test only one technique at a time, such as MaxEnt or another algorithm

known for SDM construction. Being our study area considerably wide, and as suggested by the most recent literature in ecology and geosciences, a comparative approach, based on at least two of the most advanced machine learning algorithms is essential to provide and ensure more reliable results and insights (see for example Zhao et al. 2022 [ref. 90]) to better understand niche distribution and their dynamics.

On this regard, we thank very much Reviewer #1 for pointing at the one of the main strengths underlining our work: “...*this kind of combination of data integration and modeling as an important and promising route forward for archaeology... the exploration of RF and MaxEnt is a step forward in most respects.*” and Reviewer #2 for considering that “[...]the results are of wide relevance to studies of Neolithisation processes.”

REVIEWER COMMENTS

Green : Response to reviewer comments

Reviewer #1 (Remarks to the Author):

The authors have revised the manuscript significantly, and to good effect: the study and its results are presented with much greater clarity. However, it is still quite long and discursive, more focused on describing the process of this study than making an argument based on the results.

I would suggest that for a big-data archaeological modeling paper to be effective it should do three things: 1) make the case that the data are of sufficient quality that the results should be reliable, 2) argue that the methods employed are appropriate to the questions under consideration, and 3) make a significant claim about the human past.

Following the revisions, this paper does a thorough job with (2), but doesn't really consider (1) in detail (though it is acknowledged) and never really gets around to (3) (ironically, since I think there is considerable potential to do make interesting and significant claims).

We have revised the text accordingly to make these points, which were already in the paper, more visible. Point 1 has been in the paper since its first version, at the beginning of the discussion (Lines 377 and ff).

There is a clear statement of purpose at the beginning of the Conclusion (p31): "The main objective of this research was to provide a first characterization for both the ecological and the environmental niche of Neolithic settlements and their food plant resources, cultivation practices and variations in time and space across the Western Mediterranean area, based on a big database and a data-driven approach using machine learning techniques." This *should* be (3) - but the paper never provides a simple statement of what those occupied niches are, and how they compare to the niches of particular food plants! I think that it could, and that that would provide the summary message that is not so far clearly articulated. If I follow the modeling process correctly (p25), it seems that two distinct datasets have been analyzed: one of site locations, and one of site locations where particular crops have been found. If that's the case, it seems to me that an obvious question would be how the observed niches based on site distribution for a given time period relate to the observed trans-temporal crop niches. That is, if - based on all occurrences (it sounds like the data are presence/absence rather than abundance?) of, e.g., chickpeas - it is possible to define a chickpea niche, then that niche might be compared to the one occupied by Neolithic farmers for any given time period.

We disagree with the reviewer on this point. The reviewer misunderstands one important goal of this paper. We cannot compare the human and crop niches because crop remains come from the same sites inhabited by humans and similar crops are present at most sites within the same chronological phase. What this paper intends is to model and quantify human niche breadth changes over time and compare it to climatic changes and crop changes. Clarifying this point has helped us to formulate our "significant claim" about human past in our conclusions and we thank the reviewer for insisting on it.

This would provide a means of *using* the model results. The four conclusions on p32, in contrast - while interesting and thoughtful - seem to have more to do with general observations of the datasets compiled than they do with the modeling results. That being the case, the claim that follows - "the combined analyses performed in this paper prove extremely useful to answer questions regarding agricultural decision-making and spreading in the past" - doesn't seem to follow from the paper. I don't think that this is necessarily a shortcoming of the study - it may be the inevitable outcome of uncertainty in the data and ambiguity in the results - but it is a problem that could be avoided with a

clearer focus on a few key things that the authors have learned about the Neolithic.

The text has been revised following the reviewer's comments.

Finally, landscape taphonomy (patterns of site preservation and discovery) and site taphonomy (patterns of preservation of archaeobotanical remains) are both of great importance here, as are data biases more generally. There is some explicit recognition that these may exist (e.g., p21), but not enough consideration of the ways in which these might affect the dataset, analyses, and subsequent conclusions.

We acknowledge both taphonomic factors affecting our datasets and that is why we use a ML approach for our analysis. Our region includes sites with waterlogged preservation and with charred preservation of archaeobotanical remains, an ideal scenario when compared to other study areas. We also integrate different landscapes and hence overcome local taphonomic processes by working at a broader geographical scale. We are conservative in our conclusions and do not overinterpret our results for crop species or regions where taphonomic factors are most relevant.

Specific comments on the text:

Ln66-68

“The main hypothesis of the paper is that choices regarding site location and crop cultivation made in periods of climatic change most likely reflect adaptive strategies of prehistoric populations.”

Is this the hypothesis of the paper, or an assumption underlying the analysis? This is not really a hypothesis that the paper tests, but rather an underlying premise: by looking at site locations we can garner information about adaptive strategies.

The text has been reformulated according to the reviewer's comments.

Ln77-82

Does this abundance of data on pile dwelling spatially bias the dataset towards some areas at the expense of others?

More generally, it would be important to recognize in the Introduction that an HS approach based on archaeological settlement data is vulnerable to taphonomic and recovery biases. Perhaps those don't matter here, but that's an argument that needs to be made.

The paper repeatedly makes the argument that archaeological datasets involve biases (always) and that precisely a machine learning approach is necessary. The text presents the abundance of data coming from the pile-dwelling sites as an advantage, not as a disadvantage, since we probably correct a bias (i.e. generated when not having waterlogged sites), and we also say that a lot of new data for other regions has become available thanks to the rescue archaeology, so these periods are well researched in general

Ln164-166

“These changes are unlikely to be the result of a climate change that would make these regions suddenly fall into the suitability area of these crops.”

Ok - why is that unlikely? Isn't determining that likelihood one of the objects of the modeling described here?

We have modified the text according to the rev. comment.

p6, Lns 176-181

"The aims of the paper are to 1) explore and reveal possible climatic dynamics in the study area; 2) examine the settings of human and crop niches; 3) evaluate landscape/climate-level factors that may have contributed to the crops shifting or to maintaining these distributions over space and time; 4) while mapping potential suitable areas of early farmers' settlements; 5) ultimately, to assess the capacity of these techniques to address relevant archaeological and archaeobotanical questions."

The way these aims are framed raises some questions. With respect to (1), isn't climate data an input, not an output? With respect to (2) and (3) – isn't there a more basic question here, which is simply that of determining whether the niches occupied by Neolithic farmers changed over time? From that follow others: did niches move in space following climate changes, but remain stable in character? If they changed in character, was that related to the adoption of different crops? There are lot sof interesting questions with which this research is engaged but which aren't quite articulated clearly enough here.

Item (4) is not an aim – it needs a verb, and (5) would be more effective if it were more specific about what the relevant questions were. Aim (5) also raises the question of whether the paper's ultimate goal to assess the techniques, or to say something about Neolithic farming?

Thank you for the comment. We have reformulated our questions according to these observations.

Fig. 1

This map raises the question of data biases, addressing which is, I would argue, fundamental to the project. Surely part of the story of this map is that sites are scarce in the Po Valley at least in part for reasons of landscape taphonomy. This has been the subject of some attention, e.g. recently:

Marco Cavalazzi. (2021). "Looking through the keyhole": problems and research strategies for landscape archaeology in an alluvial plain with a high rate of vertical growth. The case of Bassa Romagna and south-eastern Po Valley. GROMA: Documenting Archaeology, 5. <https://doi.org/10.32028/Groma-Issue-5-2020-1358>

That provides the opportunity to bring up the issue, and argue that it does not compromise the results. Perhaps the patterns that the authors detect are so robust that biases are irrelevant, but that's an argument that has to be made.

Fig. 2

These are, I guess, CHELSA Trace21k data spatially averaged over the study area? That should be made clear in the caption.

We have modified the Fig.2 legend

Ln198-201

"four Phases are based on crop dynamics and divide our study period according to changing

dominance of crop assemblages”

This is an interesting approach (and possibly a paper in its own right!). How does the division compare with traditional culture historical divisions?

This was the best solution given that each subregion has its own cultural historical dynamics

Fig 3

Caption mentions boxplots, but there are no boxplots in the figure.

Yes, in the fig. there are boxplots, even if they are small.

Ln265

“absence of suitable areas in the Po valley”

This is why the potential biases in the data are an issue. If there are sites in the Po Valley but they’re underrepresented, then those areas will necessarily be classified as low suitability.

We do not completely understand that this statement is pointed out here. This is not a general statement for the Po valley. For some periods, we have high suitability. Taphonomy certainly affects the area and this is acknowledged in our paper and by other authors we are citing but there is no reason to think that the absence of suitable areas in only one particular phase of the Neolithic is due to taphonomic reasons.

Ln271-273

Sentence fragment. And an interesting claim - why not?

Thank you for the comment. We have reformulated the sentence to avoid any misunderstanding.

Ln334-336

If I understand correctly, this is based on presence of particular crops at various sites/phases. That being the case - different issues of bias/taphonomy to site preservation/dating need to be considered (as indeed is noted below on p21).

We provided more details about the crops dataset in the introduction.

Fig.7

A qualitative color palette would be more appropriate than a ramp here, since these are discrete values rather than a continuum.

The choice of the ramp palette here is made to respect a visual identity through the entire manuscript and the supplementary material (aesthetic reasons). However, bar labels have been added within the fig. for more clarity.

p22

“...the distribution of Neolithic settlements and the main cultivated crops needed to overcome different adaptive challenges in different zones of the Western Mediterranean area and that these challenges varied over time.”

This might be true – and is an interesting claim – but doesn’t seem to me like a direct outcome of the model. I think it would be more appropriate as an interpretation to be argued.

We modified the text accordingly.

Ln449-453

“With some exception, the differences in climatic niches for barley, glume wheats and naked wheats are less relevant, which could point at farmer’s choices and cultural influences as determining factors rather than ecological ones, at least when considering broad patterns (local or regional studies might be more suited to identify climate-related agricultural decision-making).”

Unclear. Why less relevant?

This paragraph has been deleted. Indeed, it was unclear.

Ln631

“Numbtechnique”?

Corrected

Ln669 & 674

“predominant taxa” and “gain importance”

Maybe this is an outcome of the compilation of the archaeobotanical database that’s described above, but is not so far as I can tell an outcome of the modeling. In fact I don’t think there’s been any discussion of the relative abundance of different taxa in time/space.

Figure 7 and from lines 402 and ff. discuss the predominance of cereals through time.

Ln679-680

“farmers responded to climate change by moving to higher altitude (broadening their niche)”

Is this niche expansion associated with population growth? Of course population proxies present a difficult problem in their own right, and one perhaps beyond the scope of this study, but there’s an important question here that gets at the issue of adaptation on which the authors focus: are farmers moving in order to continue occupying the same niche space as climate change causes shifts in niche locations (i.e. mobility as adaptation), or are farmers moving for other reasons (e.g. population growth) and necessarily having to adapt to the new niches in which they find themselves? These are

very different scenarios, and to my mind demonstrate that simply appealing to “adaptation” is too general.

We have reformulated the text to make our conclusions more incisive. Unfortunately the radiocarbon data is not sufficiently systematic for our study area to allow further analyses on population growth as suggested by the reviewer here.

Reviewer #3 (Remarks to the Author):

I thanks the authors for extensively rewriting the text and answering all my queries and those from the other reviewers

Reviewer #3 (Remarks on code availability):

Code not available - unless mistake of mine - in the package made available to the reviewers, and not publicly available yet on the github repository linked in the manuscript

We attached the latest code to the submission documents. Code will be publicly available on GitHub once the paper is accepted